# An atlas of the aging lung mapped by single cell transcriptomics and deep tissue proteomics

Ilias Angelidis[1], Lukas M. Simon [2], Isis E. Fernandez[1], Maximilian Strunz[1], Christoph H. Mayr[1], Flavia R. Greiffo[1], George Tsitsiridis[2], Meshal Ansari[1,2], Elisabeth Graf[3], Tim-Matthias Strom[3], Monica Nagendran[4], Tushar Desai [4], Oliver Eickelberg [5], Matthias Mann [6], Fabian J. Theis [2,7] & Herbert B. Schiller[1]

Aging promotes lung function decline and susceptibility to chronic lung diseases, which are the third leading cause of death worldwide. Here, we use single cell transcriptomics and mass spectrometry-based proteomics to quantify changes in cellular activity states across 30 cell types and chart the lung proteome of young and old mice. We show that aging leads to increased transcriptional noise, indicating deregulated epigenetic control. We observe cell type-specific effects of aging, uncovering increased cholesterol biosynthesis in type-2 pneumocytes and lipofibroblasts and altered relative frequency of airway epithelial cells as hallmarks of lung aging. Proteomic profiling reveals extracellular matrix remodeling in old mice, including increased collagen IV and XVI and decreased Fraser syndrome complex proteins and collagen XIV. Computational integration of the aging proteome with the single cell transcriptomes predicts the cellular source of regulated proteins and creates an unbiased reference map of the aging lung.

[1] Helmholtz Zentrum München, Institute of Lung Biology and Disease, Member of the German Center for Lung Research (DZL), Munich 85764, Germany. [2] Helmholtz Zentrum München, Institute of Computational Biology, Munich 85764, Germany. [3] Helmholtz Zentrum München, Institute of Human Genetics, Munich 85764, Germany. [4] Department of Internal Medicine, Division of Pulmonary and Critical Care, Institute for Stem Cell Biology and Regenerative Medicine, Stanford University School of Medicine, Stanford 94305 CA, USA. [5] Department of Medicine, Division of Respiratory Sciences and Critical Care Medicine, University of Colorado, Aurora 80045 CO, USA. [6] Department of Proteomics and Signal Transduction, Max Planck Institute of Biochemistry, Martinsried, Munich 82152, Germany. [7] Department of Mathematics, Technische Universität München, Munich 85748, Germany. These authors contributed equally: Ilias Angelidis, Lukas M. Simon. Correspondence and requests for materials should be addressed to F.J.T. (email: fabian.theis@helmholtz-muenchen.de) or to H.B.S. (email: herbert.schiller@helmholtz-muenchen.de)

The intricate structure of the lung enables gas exchange between inhaled air and circulating blood. As the organ with the largest surface area (~70 m² in humans), the lung is constantly exposed to a plethora of environmental insults. A range of protection mechanisms are in place, including a highly specialized set of lung-resident innate and adaptive immune cells that fight off infection, as well as several stem and progenitor cell populations that provide the lung with a remarkable regenerative capacity upon injury[1]. These protection mechanisms seem to deteriorate with advanced age, since aging is the main risk factor for developing chronic lung diseases, including chronic obstructive pulmonary disease (COPD), lung cancer, and interstitial lung disease[2,3]. Advanced age causes a progressive impairment of lung function even in otherwise healthy individuals, featuring structural and immunological alterations that affect gas exchange and susceptibility to disease[4]. Aging decreases ciliary beat frequency in mice, thereby decreasing mucociliary clearance and partially explaining the predisposition of the elderly to pneumonia[5]. Senescence of the immune system in the elderly has been linked to a phenomenon called 'inflammaging', which refers to elevated levels of tissue and circulating pro-inflammatory cytokines in the absence of an immunological threat[6]. Several previous studies analyzing the effect of aging on pulmonary immunity point to age-dependent changes of the immune repertoire as well as activity and recruitment of immune cells upon infection and injury[4]. Vulnerability to oxidative stress, pathological nitric oxide signaling, and deficient recruitment of endothelial stem cell precursors have been described for the aged pulmonary vasculature[7]. The extracellular matrix (ECM) of old lungs features changes in tensile strength and elasticity, which were discussed to be a possible consequence of fibroblast senescence[8]. Using atomic force microscopy, age-related increases in stiffness of parenchymal and vessel compartments were demonstrated recently[9]; however, the causal molecular changes underlying these effects are unknown.

Aging is a multifactorial process that leads to these molecular and cellular changes in a complicated series of events. The hallmarks of aging encompass cell-intrinsic effects, such as genomic instability, telomere attrition, epigenetic alterations, loss of proteostasis, deregulated nutrient sensing, mitochondrial dysfunction, and senescence, as well as cell-extrinsic effects, such as altered intercellular communication and extracellular matrix remodeling[2,3]. The lung contains potentially at least 40 distinct cell types[10], and specific effects of age on cell-type level have never been systematically analyzed.

In this study, we build on rapid progress in single-cell transcriptomics[11,12] which recently enabled the generation of a first cell-type resolved census of murine lungs[13], serving as a starting point for investigating the lung in distinct biological conditions as shown for lung aging in the present work. We computationally integrate single-cell signatures of aging with state-of-the-art whole lung RNA-sequencing (RNA-seq) and mass spectrometry-driven proteomics[14] to generate a multi-omics whole organ resource of aging-associated molecular and cellular alterations in the lung.

## Results

**Lung aging atlas reveals deregulated transcriptional control**. To generate a cell-type resolved map of lung aging we performed highly parallel genome-wide expression profiling of individual cells using the Dropseq workflow[15] which uses both molecule and cell-specific barcoding, enabling great cost efficiency and accurate quantification of transcripts without amplification bias[16]. Single-cell suspensions of whole lungs were generated from 3-month-old mice (n = 8) and 24-month-old mice (n = 7). After quality

control, a total of 14,813 cells (7672 young, 7141 old) were used for downstream analysis (Fig. 1a). Quality metrics including number of unique molecular identifiers (UMI), genes detected per cell, and reads aligned to the mouse genome were comparable across mice (Supplementary Fig. 1a–c). To ensure that cell-type discovery is not confounded by aging effects, we only used highly variable genes between cell types (see Methods for details). Unsupervised clustering analysis revealed 36 distinct clusters corresponding to 30 cell types, including all major known epithelial, mesenchymal, and leukocyte lineages (Fig. 1b, c). We observed very good overlap across mouse samples (Silhouette coefficient: −0.074) and most clusters were derived from >70% of the mice of both age groups (Supplementary Fig. 1d and e). The definition of cell types (clusters in t-distributed stochastic neighbor embedding (tSNE) map) was very comparable between old and young mice, indicating that the cell-type identity was not strongly confounded by the aging effects (Supplementary Fig. 1f). Two clusters exclusively contained cells from a single mouse and were removed from downstream analysis. Interestingly, we identified even rare (<1%, 43 cells) cell types such as megakaryocytes, which were recently identified as an unexpected tissue-resident cell type in mouse lung[17]. Of note, some samples contributed as little as a single cell to this megakaryocyte cluster, emphasizing the power and accuracy of the computational workflow used here for data integration from multiple mice.

We used differential gene expression analysis to determine cell type-specific marker genes with highly different levels between clusters (Fig. 1c, Supplementary Data 1). The clusters were annotated with assumed cell-type identities based on (1) known marker genes derived from expert annotation in literature and (2) enrichment analysis using Fisher's exact test of gene expression signatures of isolated cell types from databases including ImmGen[18] and xCell[19]. Correlation analysis of marker gene signatures revealed that similar cell types clustered together, implying correct cell-type annotation (Fig. 1c).

We used the matchSCore tool[20] to compare the cluster identities of our dataset with the lung data in the recently published Mouse Cell Atlas (MCA)[13], and found very good agreement in cluster identities and annotations (Supplementary Fig. 2a). Moreover, when comparing our cluster identities to the MCA peripheral blood data, only weak correspondence was observed (Supplementary Fig. 2b), which was similar in the MCA peripheral blood versus MCA lung comparison (Supplementary Fig. 2c). One notable exception in this comparison is the cluster of red blood cells in our dataset which achieved high correspondence with the MCA peripheral blood cluster annotated as Erythroblast_Hbb-a2_high. The red blood cells serve as a control and illustrate matchSC values for a correct overlap (Supplementary Fig. 2d). Taken together, these findings indicate that very little blood-derived contamination was present.

Additionally, we noticed one cluster of mainly proliferating cells showing high expression levels for S and G2M cell-cycle marker genes (Supplementary Fig. 3a and b). Young mice showed a higher fraction of cells in this cluster compared to old mice (Supplementary Fig. 3c; Generalized linear binomial model, p < 0.001). Next, we isolated this cluster and corrected the gene expression levels for cell-cycle phase (Supplementary Fig. 3 d and e). Subsequent unsupervised clustering analysis revealed that these proliferating cells belong to T cells, type-2 pneumocytes, and alveolar macrophages (Supplementary Fig. 3f–i).

It was suggested that aging is a consequence of increased transcriptional instability rather than the result of a coordinated transcriptional program, and that an aging-associated increase in transcriptional noise can lead to fate drifts and ambiguous cell-type identities[21,22]. Therefore, we quantified transcriptional noise following previous work[22] and accounted for differences in total

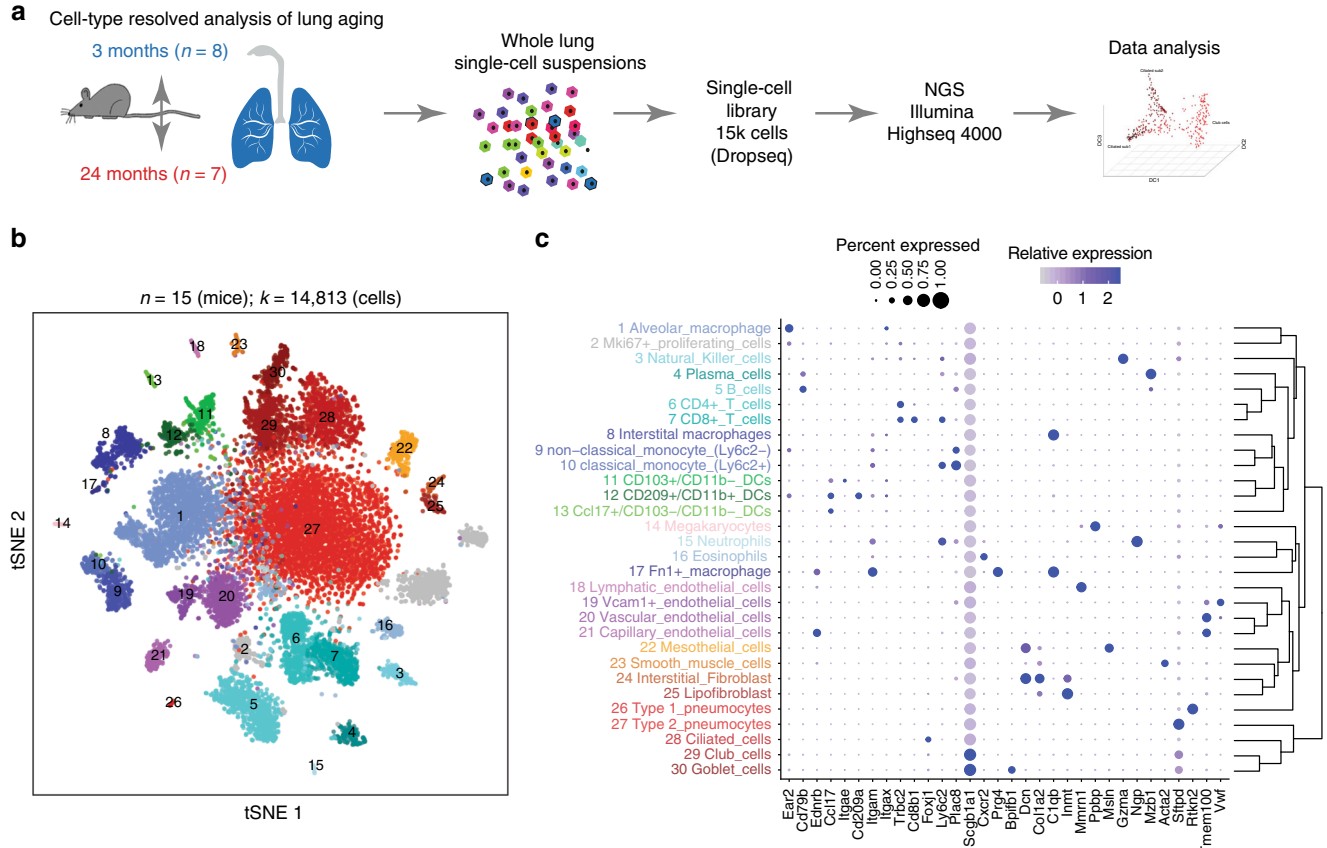

**Fig. 1** A single-cell atlas of mouse lung reveals major cell-type identities. **a** Experimental design—whole lung single-cell suspensions of young and old mice were analyzed using the Dropseq workflow. **b** The *t*-distributed stochastic neighbor embedding (tSNE) visualization shows unsupervised transcriptome clustering, revealing 30 distinct cellular identities. **c** The dotplot shows (1) the percentage of cells expressing the respective selected marker gene using dot size and (2) the average expression level of that gene based on unique molecular identifier (UMI) counts. Rows represent hierarchically clustered cell types, demonstrating similarities of transcriptional profiles

UMI counts and cell-type frequencies (see Methods for details). We observed an increase in transcriptional noise with aging in most cell types (Fig. 2a). To further exclude technical confounding we additionally averaged the transcriptional noise scores per mouse and obtained highly concordant results (Fig. 2b). To further substantiate this finding we quantified transcriptional noise in an alternative manner using Spearman's correlations between cells. This analysis confirmed our finding that transcriptional noise is increased with aging (Fig. 2c, d) and is in line with previous reports in the human pancreas[22] or mouse CD4+ T cells[21].

**Multi-omics data integration of mRNA and protein**. To validate the completeness of our single-cell RNA-sequencing (scRNA-seq) data and capture age-dependent alterations in both mRNA and protein content for the whole lung, we generated two additional cohorts of young and old mice (Fig. 3a, Supplementary Figure 4 and Supplementary Data 2): (1) bulk RNA-seq data of three replicates of young (3 months) and old mice (22 months) and (2) state-of-the-art shotgun proteomics data of four replicates of young (3 months) and old mice (24 months). To compare the whole lung bulk transcriptome with single-cell data we generated in silico bulk samples from the scRNA-seq data by summing expression counts from all cells for each mouse individually (Supplementary Data 2). Differential gene expression analysis from in silico bulks and real whole lung bulk sequencing revealed a total of 2362 and 9245 differentially expressed genes (negative

binomial generalized linear model, false discovery rate (FDR) <10%) between the two age groups, respectively (Supplementary Fig. 4a, b, Supplementary Data 2). From whole lung tissue proteomes we quantified 5212 proteins across conditions and found 213 proteins to be significantly regulated with age (two-sided *t*-test, FDR < 10%, Supplementary Fig. 4c, Supplementary Data 2). We observed very good agreement between the real and in silico bulk data, thus excluding strong biases by the single-cell isolation procedures (Fig. 3b). Furthermore, we also observed strong correspondence between the age-dependent alterations in all three data sets (Fig. 3c), indicating that we were able to pick up robust age-dependent changes with three independent experimental settings. Significant correlation was observed between the gene-level fold changes derived from RNA-seq, scRNA-seq, and protein expression data (Supplementary Fig. 4d–f).

Prediction of the upstream regulators[23] of the observed expression changes in either the transcriptome or proteome data gave very similar results (Fig. 2d). In both datasets from independent mouse cohorts, we discovered a pro-inflammatory signature, which included upregulation of *Il6*, *Il1b*, *Tnf*, and *Ifng*, as well as the downregulation of *Pparg* and *Il10* (Fig. 2d). Furthermore, to reveal common or distinct regulation of gene annotation categories in the transcriptome or proteome, we performed a two-dimensional annotation enrichment analysis[24] (Supplementary Data 3). Again, most gene categories regulated by age were showing the same direction in transcriptome and proteome so that the positive Pearson's correlation of the annotation enrichment scores was highly significant (Fig. 2e).

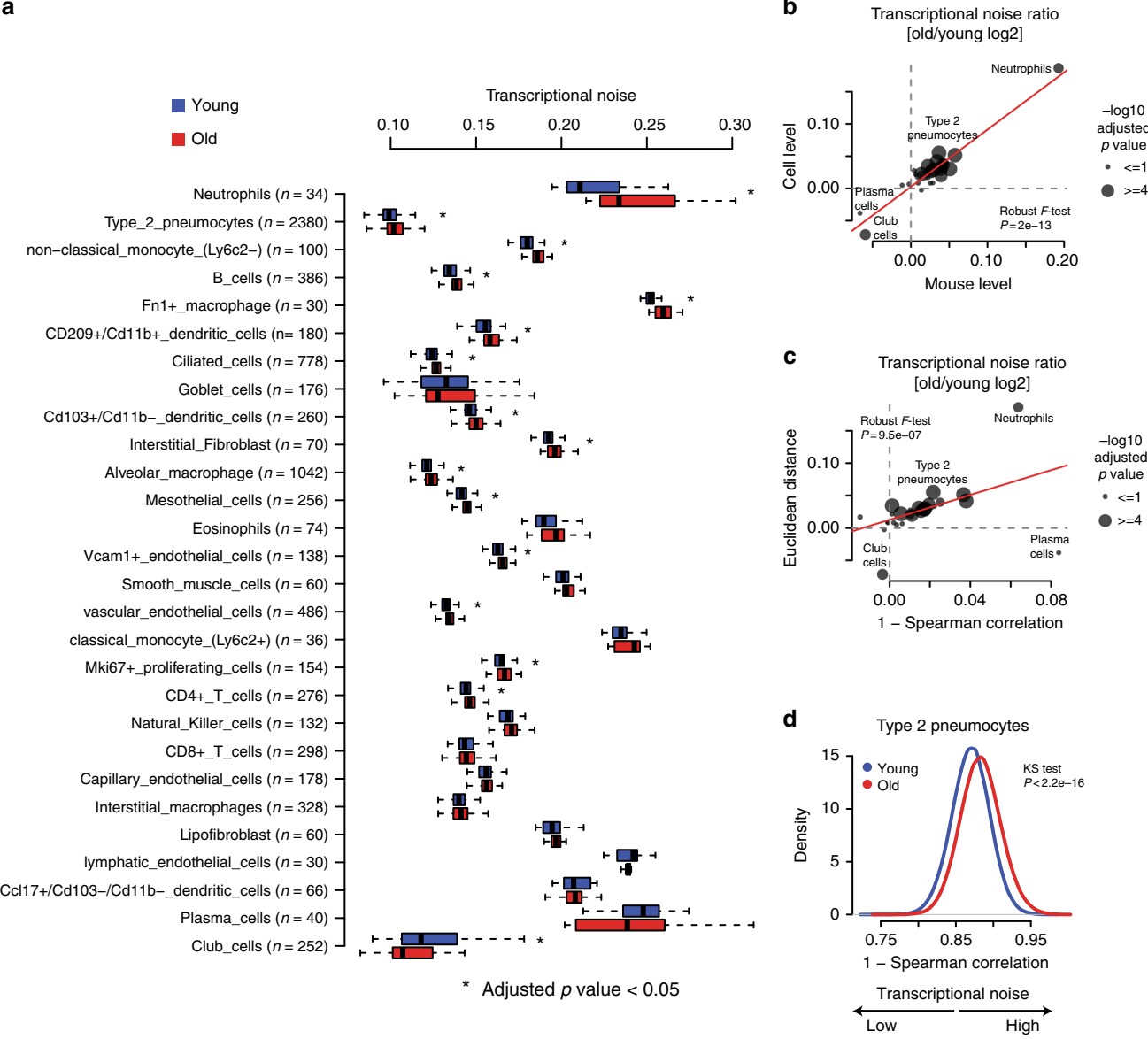

**Fig. 2** Most cell types show increased transcriptional noise with aging. **a** Boxplot illustrates transcriptional noise by age and cell type for the indicated number of cells. For all boxplots, the box represents the interquartile range, the horizontal line in the box is the median, and the whiskers represent 1.5 times the interquartile range. Blue and red colors indicate young and old cells, respectively. Asterisk indicates significant changes (Wilcoxon's rank sum test, adjusted $p$ value < 0.05). Cell types are ordered by decreasing transcriptional noise ratio between old and young cells. **b** Scatterplot shows the log2 ratio of transcriptional noise between old and young samples as calculated using mouse averages ($n = 15$) and single cells on the X and Y axes, respectively. **c** Scatterplot depicts the log2 ratio of transcriptional noise between old and young samples as calculated using 1–Spearman correlation and the Euclidean distance between cells on the X and Y axes, respectively. For both panels, the size of the dots corresponds to the negative log10 adjusted $p$ value of the cell type-resolved differential transcriptional noise test and the red lines correspond to the robust linear model regression fit. **d** As an example, the distribution of 1–Spearman correlation coefficients between all pairs of young and old cells is shown for type-2 pneumocytes. Larger values represent increased transcriptional noise. Blue and red colors indicate young and old samples

We observed several hallmarks of aging, including a decline in mitochondrial function and upregulation of pro-inflammatory pathways ('inflammaging'). Interestingly, we detected a strong increase in immunoglobulins in both datasets, as well as higher levels of major histocompatibility complex (MHC) class I, which is consistent with the observed increase in the interferon pathway (Fig. 2e). Many extracellular matrix genes, such as collagen III, were downregulated on both the mRNA and protein levels, while the levels of all basement membrane-associated collagen IV genes were increased on the protein level, but decreased at the mRNA level in both transcriptome datasets (Fig. 2f) and in proximity

ligation in situ hybridization of mRNA in tissue sections (Fig. 2g). The differential regulation of collagen IV transcripts and proteins highlights the importance of combined RNA and protein analysis. We validated the increased protein abundance of collagen IV using immunofluorescence and found that interestingly the main increase in collagen IV in old mice was found around airways and vessels (Fig. 2g).

**Altered frequency of airway epithelial cells upon aging**. Single-cell RNA-seq can disentangle relative frequency changes of cell types from real changes in gene expression within a given cell

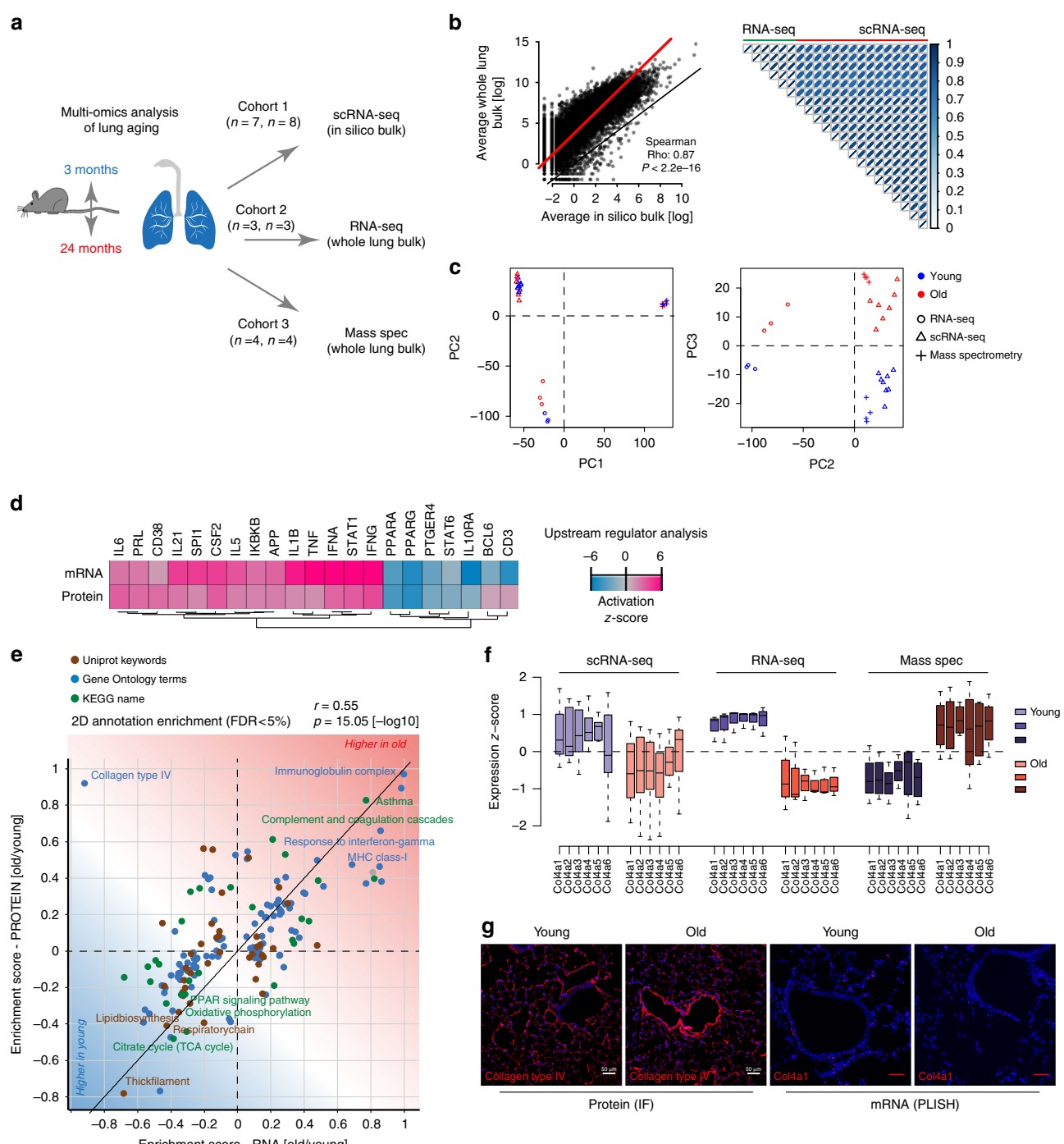

type. We analyzed age-dependent alterations of relative frequencies of the 30 cell types represented in our dataset. Since the cell-type frequencies are proportions, the data are compositional. Therefore, it is impossible to statistically discern if a relative change in cell-type frequency is caused by the increase of a given cell type or the decrease of another. However, after performing dimension reduction using multidimensional scaling of the cell-type proportions, we observed a significant association between the first coordinate and age (Fig. 4a, b; Wilcoxon test, $p < 0.005$), indicating that cell-type frequencies differed between young and old mice. Interestingly, the Dropseq data showed a relative increase in ciliated cells in old mice so that the ratio of club to ciliated cells was altered (Fig. 4c, d). Relative frequency differences in scRNA-seq data can be biased by tissue isolation

artifacts. We therefore validated the change in club to ciliated cell proportions by deconvolving the whole lung bulk expression data using our single-cell gene expression profiles (Fig. 4e). Indeed, we found that the ciliated cell marker genes signature was significantly upregulated in old compared to young mouse lungs (Fig. 4f). Interestingly, this analysis also revealed marked increase of various immune cell populations, including CD4+ and CD8+ T cells, eosinophils, and classical monocytes (Fig. 4e). We additionally validated this finding in situ by quantifying airway club and ciliated cells using immunostainings of Foxj1 (ciliated cell marker) and CC10 (club cell marker) (Fig. 4g). In addition, in this analysis the ciliated cells were increased in old mice (Fig. 4h), leading to a significantly altered ratio of club to ciliated cells in aged mouse airways (Fig. 4i).

**Fig. 3** Multi-omic data integration uncovers uncoupled regulation of RNA and protein. **a** Experimental design—three independent cohorts of young and old mice were analyzed by single-cell RNA-sequencing (scRNA-seq), bulk RNA-seq, and mass spectrometry-driven proteomics respectively. **b** On the left, gene expression profiles from whole lung bulk samples ($n = 6$) and in silico bulk samples ($n = 15$) were averaged and plotted on $X$ and $Y$ axes, respectively. Red and black lines indicate linear model fit and the diagonal. On the right, correlation heatmap shows Pearson's correlation between all bulk and in silico bulk samples. **c** Normalized bulk (RNA-seq) and in silico bulk (scRNA-seq) data were merged with proteome data (mass spectrometry) and quantile normalized. The first two principal components show clustering by data modality. The third principal component separates young from old samples across all three data modalities. Blue and red colors indicate young and old samples, respectively. **d** Gene expression and protein abundance fold changes were used to predict upstream regulators that are known to drive gene expression responses similar to the ones experimentally observed. Upstream regulators could be cytokines or transcription factors. The color-coded activation $z$-score illustrates the prediction of increased or decreased activity upon aging. **e** The scatter plot shows the result of a two-dimensional annotation enrichment analysis based on fold changes in the transcriptome ($x$-axis) and proteome ($y$-axis), which resulted in a significant positive correlation of both datasets. Types of databases used for gene annotation are color coded as depicted in the legend. **f** Expression of collagen IV genes in the in silico bulk (scRNA-seq), bulk (RNA-seq), and proteome (mass spec) experiments, respectively. The box represents the interquartile range, the horizontal line in the box is the median, and the whiskers represent 1.5 times the interquartile range. **g** Immunofluorescence image of collagen type IV using confocal microscopy at ×25 magnification and proximity ligation in situ hybridization (PLISH) staining of *Col4a1* mRNA. Note the increased fluorescence intensity of the protein around vessels in old mice along with the decreased mRNA expression (scale bar: 50 μm)

**Altered composition of the pulmonary extracellular matrix**. The ECM can act as a solid phase-binding interface for hundreds of secreted proteins, creating an information-rich signaling template for cell function and differentiation[25]. Alterations in ECM composition and possibly architecture in the aging lung have been suggested[26], but experimental evidence using unbiased mass spectrometry is scarce. From the 5138 proteins quantified in the tissue proteome (Fig. 5a), we identified 32 Matrisome proteins with significant change upon aging (two-sided $t$-test, FDR < 10%, Fig. 5b, Supplementary Data 2). Collagen XIV, a collagen of the FACIT (Fibril Associated Collagens with Interrupted Triple helices) family of collagens that is associated with the surface of collagen I fibrils and may function by integrating collagen bundles[27], was downregulated in old mice (Fig. 5b). Collagen XIV is a major ECM binding site for the proteoglycan Decorin[28], which is known to regulate TGF-beta activity[29,30]. Interestingly, our scRNA-seq data localized collagen XIV expression to interstitial fibroblasts that together with mesothelial cells also expressed Decorin and were distinct from the lipofibroblasts that showed very little expression of this particular collagen (Fig. 5c). Thus, the combination of tissue proteomics with single-cell transcriptomics enabled us to predict the cellular source of the regulated proteins, which can be explored in the online webtool (https://theislab. github.io/LungAgingAtlas). In the webtool the cell-type specificity of any gene query can be exported as dot plot in pdf format.

We previously developed the quantitative detergent solubility profiling (QDSP) method to add an additional dimension of protein solubility to tissue proteomes[31–33]. In QDSP, proteins are extracted from tissue homogenates with increasing stringency of detergents, which typically leaves ECM proteins enriched in the insoluble last fraction. This enables better coverage of ECM proteins and analysis of the strength of their associations with higher-order ECM structures such as microfibrils or collagen networks. We applied this method to young and old mice and compared protein solubility profiles between the two groups (Fig. 6a). Differential comparison of the solubility profiles between young and old mice revealed 74 proteins, including 8 ECM proteins, with altered solubility profiles (two-way analysis of variance (ANOVA), FDR < 20%) (Supplementary Data 4).

Using principal component analysis of 432 secreted extracellular proteins we found that the protein solubility fractions separated in component 1, while the age groups separated in component 4 of the data (Fig. 6b). Thus, principal component analysis enabled the stratification of secreted proteins by their biochemical solubility and their differential behavior upon aging (Fig. 6c). This analysis also showed that neither the abundance nor the solubility of many ECM proteins, including collagen I and

basement membrane laminins, was altered (Fig. 6c). While the most abundant basement membrane laminin chain (Lamc1) was unaltered in both abundance (Fig. 6d) and solubility (Fig. 6g), serving as a control for overall integrity of the basement membrane and the quality of our data, the basement membrane-associated trimeric Fraser Syndrome complex (consisting of Fras1, Frem1, and Frem2) was downregulated (Fig. 6e) and more soluble (Fig. 6h) in old age. Incorporation of the Fraser syndrome complex within the basement membrane (rendering it more insoluble) has been shown to depend on extracellular assembly of all three proteins[34], indicating that this assembly and/or the expression of either one or all subunits of the complex is perturbed in old mice. Fraser syndrome is a skin-blistering disease which points to an important function of the Fraser syndrome complex proteins in linking the epithelial basement membrane to the underlying mesenchyme[34]. In the lungs of adult mice, expression is restricted to the mesothelium; Fras1−/− mice develop lung lobulation defects[35]. Interestingly, the solubility of the downregulated collagen XIV (Fig. 6f) was also significantly changed (Fig. 6i).

**Cell type-specific effects of aging**. Cell type-resolved differential gene expression testing between age groups in the single-cell data sets identified 391 significantly regulated genes (Wilcoxon rank sum test, FDR < 10%) (Fig. 7a; Supplementary Data 5). Alveolar macrophages and type-2 pneumocytes, the two cell types with highest number of cells in the dataset, are discussed as an example for the type of insight that can be gained from our cell type-resolved resource. Both cell types showed a clearly altered phenotype in aged mice.

In alveolar macrophages, we found 125 significantly regulated mRNAs (FDR < 10%, Fig. 7b), including the downregulation of the genes for Eosinophil cationic protein 1 & 2 (*Ear1* and *Ear2*), which have ribonuclease activity and are thought to have potent innate immune functions as antiviral factors[36]. We observed higher levels of the C/EBP beta (*Cebpb*), which is an important transcription factor regulating the expression of genes involved in immune and inflammatory responses[37,38]. Several genes that have been shown to be upregulated in lung injury, repair, and fibrosis[33], such as *Spp1*, *Gpnmb*, and *Mfge8*, were also induced in alveolar macrophages of old mice, which may be a consequence of the ongoing 'inflammaging'.

In alveolar type-2 pneumocytes, 121 mRNAs were significantly regulated (Wilcoxon rank sum test, FDR < 10%, Fig. 7c). We observed a strong increase of the MHC class I genes *H2-K1*, *H2-Q7*, *H2-D1*, and *B2m* (Fig. 7c), which we validated using an

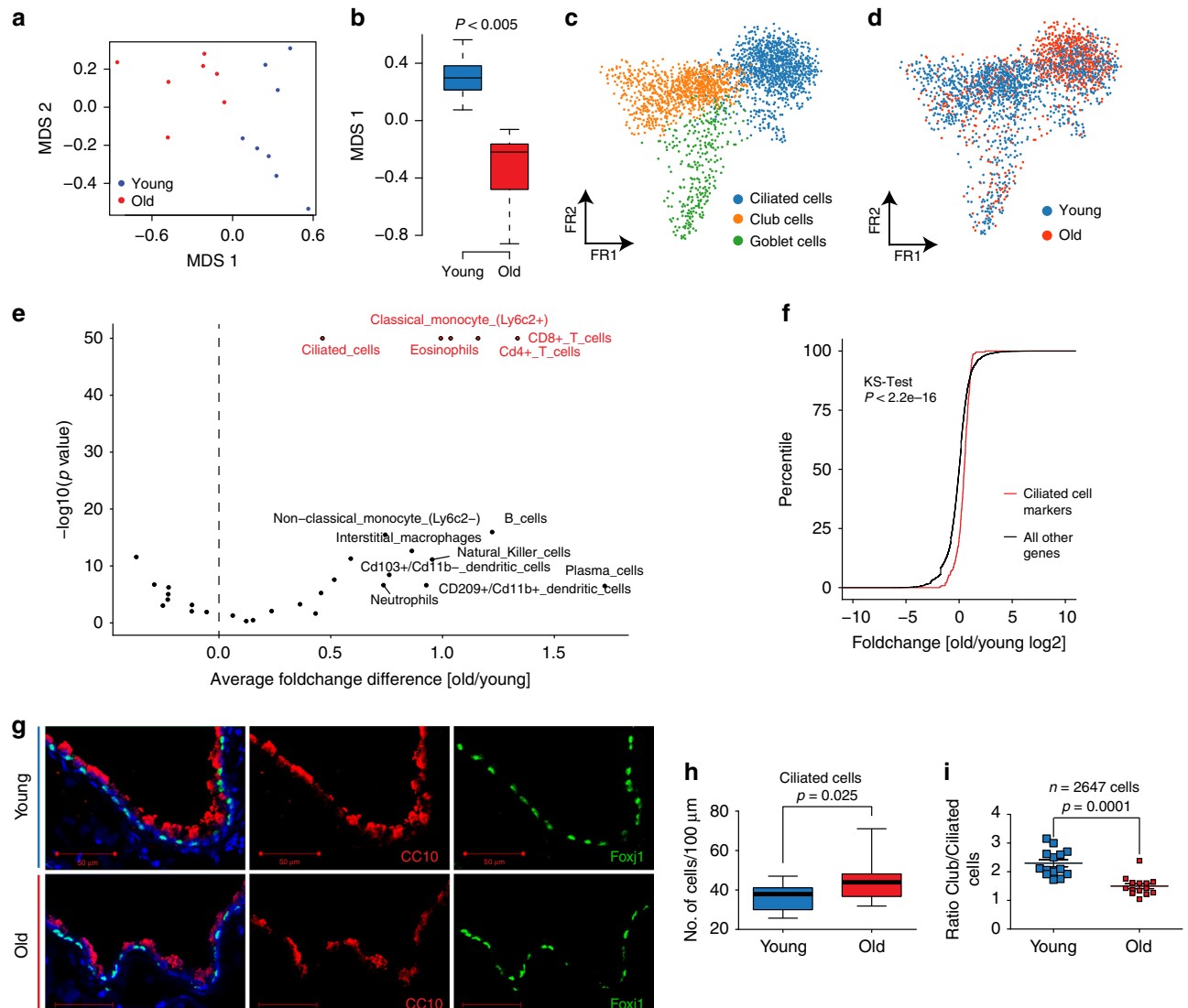

**Fig. 4** Cell-type deconvolution reveals increase of ciliated cells in airways of old mice. **a** The multidimensional scaling (MDS) plot shows the mouse-wise euclidean distances of cell-type proportions for the two age groups. **b** The box plot shows the significant difference in the multidimensional scaling component 1 of cell-type proportions between young ($n = 8$) and old ($n = 7$). The box represents the interquartile range, the horizontal line in the box is the median, and the whiskers represent 1.5 times the interquartile range. **c** The Fruchterman–Reingold (FR) embedding of the airway epithelial cells in the dataset reveals distinct clusters of airway cell identity. **d** The indicated color code shows the distribution of young and old cells to the three clusters presented in (**c**). Note the increased density of old cells in the ciliated cell cluster. **e** The volcano plot shows negative log10 enrichment *p* values of cell-type marker signatures in the differential expression results of the bulk RNA-seq data from young and old mice. **f** The empirical density plot shows significant enrichment for ciliated cell-type marker genes (red line) compared to all other genes (black line) in the distribution of fold changes derived from the bulk differential expression analysis. **g** Club and ciliated cells were stained using a CC10 and Foxj1 antibody respectively (scale bar: 50 μm). **h** The boxplot depicts the quantification of ciliated cells from counting a total of 2647 club and ciliated cells in 14 individual airways of $n = 2$ mice of each indicated age group. **i** Ratio of ciliated to club cells in 14 individual airways of two mice for each indicated age group. The *p* values are derived from an unpaired, two-tailed *t*-test using Welch's correction

independent flow cytometry experiment on epithelial cells marked by Epcam expression (Fig. 7k, l). Elevated MHC class I levels likely result in increased presentation of self-antigens to the immune system and are consistent with our observation of a prominent interferon-gamma signature in old mice (Fig. 3d), which is known to activate MHC class I expression[39]. Type-2 pneumocytes of old mice featured a highly significant upregulation of the enzyme Acyl-CoA desaturase 1 (*Scd1*), which is the fatty acyl Δ9-desaturating enzyme that converts saturated fatty acids into monounsaturated fatty acids (Fig. 7c). The age-dependent upregulation of *Scd1* in type-2 pneumocytes may have important implications since *Scd1* is thought to induce adaptive

stress signaling that maintains cellular persistence and fosters survival and cellular functionality under distinct pathological conditions[40].

To perform global validation of the cell type-resolved differential gene expression analysis for a large number of genes we flow-sorted epithelial cells and macrophages from an additional cohort of young and old mice (see Supplementary Fig. 5 for gating strategy) and performed bulk RNA-seq on these isolated cell types from young ($n = 4$) and old ($n = 4$) mice. Principal component analysis (PCA) was performed using the scRNA-seq-derived signatures of alveolar macrophages and type-2 pneumocytes. Gene expression profiles of flow-sorted epithelial

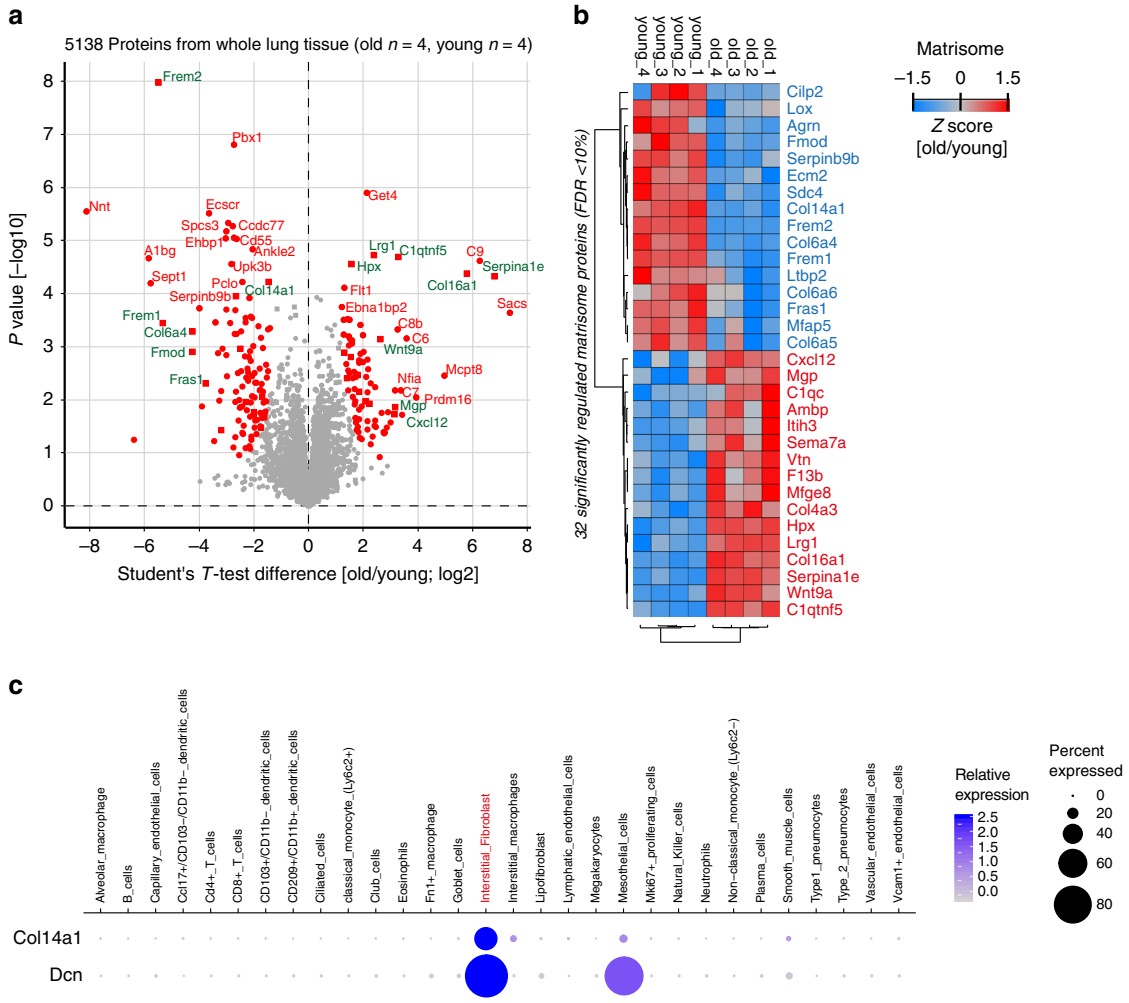

**Fig. 5** Single-cell RNA-sequencing (scRNA-seq) predicts cellular origin of age-dependent protein alterations. **a** Proteins regulated with a false discovery rate < 10% are highlighted in red in the volcano plot showing the indicated fold changes and p values derived from t-test statistic. Matrisome proteins are labeled with green gene names. **b** The z-score values of 32 significantly regulated extracellular matrix proteins were grouped by unsupervised hierarchical clustering (Pearson's correlation). **c** The dot plot shows mRNA expression specificity of *Col14a1* and its binding partner Decorin (*Dcn*) in the scRNA-seq data

cells and macrophages were projected into this PCA space (see Methods for details) showing good overlap of cell-type identity, thereby confirming the scRNA-seq-based cell-type annotation (Fig. 7d, e). Next, age-dependent alterations in the flow-sorted bulk RNA-seq data were identified (Supplementary Data 2). Significant agreement with the scRNA-seq-derived results was observed (Fisher's exact test, p < 2.2e−16, Fig. 7f–j), thus validating the power of scRNA-seq to derive age-dependent changes in gene expression.

To obtain a meta-analysis of changes in previously characterized gene expression modules and pathways, we used cell type-resolved mRNA fold changes for gene annotation enrichment analysis (Supplementary Fig. 6a and b, Supplementary Data 6) and upstream regulator analysis (Supplementary Fig. 6c–e). The analysis revealed cell type-specific alterations in gene expression programs upon aging. For instance, comparing club cells to type-2 pneumocytes showed that *Nrf2* (*Nfe2l2*)-mediated oxidative stress responses were higher in type-2 pneumocytes of old mice and lower in club cells (Supplementary Fig. 6c). Aging is known to affect growth signaling via the evolutionary conserved Igf-1/Akt/mTOR axis. Interestingly, we found evidence for increased mammalian target of rapamycin (mTOR) signaling in type-2 and club cells, but not in ciliated and goblet cells (Supplementary

Fig. 6c). Mesenchymal cells showed remarkable differences in their aging response (Supplementary Fig. 6d). For instance, we observed the pro-inflammatory Il1b signature in capillary endothelial cells, as well as in mesothelial and smooth muscle cells, but not in the other mesenchymal cell types. In myeloid cell types we found both differences and similarities in the aging response (Supplementary Fig. 6e). While an increased interferon-gamma and reduced Il10 signature in old mice was consistently observed, other effects were more specific, such as the increase in *Stat1* target genes in classical monocytes (*Ly6c2*+), which was not observed in non-classical monocytes (*Ly6c2*−).

**Increased cholesterol biosynthesis in aged cell types.** Pulmonary surfactant homeostasis is a tightly regulated process that involves synthesis of lipids by type-2 pneumocytes and lipofibroblasts[41]. Lipid metabolism in alveolar type-2 cells is regulated by sterol-response element-binding proteins (SREBPs) such as *Srebf2* and their negative regulators *Insig1* and *Insig2*. Deletion of *Insig1/2* in mouse type-2 pneumocytes activated SREBPs and led to the accumulation of neutral lipids (cholesterol esters and trigylcerids) in type-2 pneumocytes and alveolar macrophages, accompanied by lipotoxicity-related lung inflammation and tissue

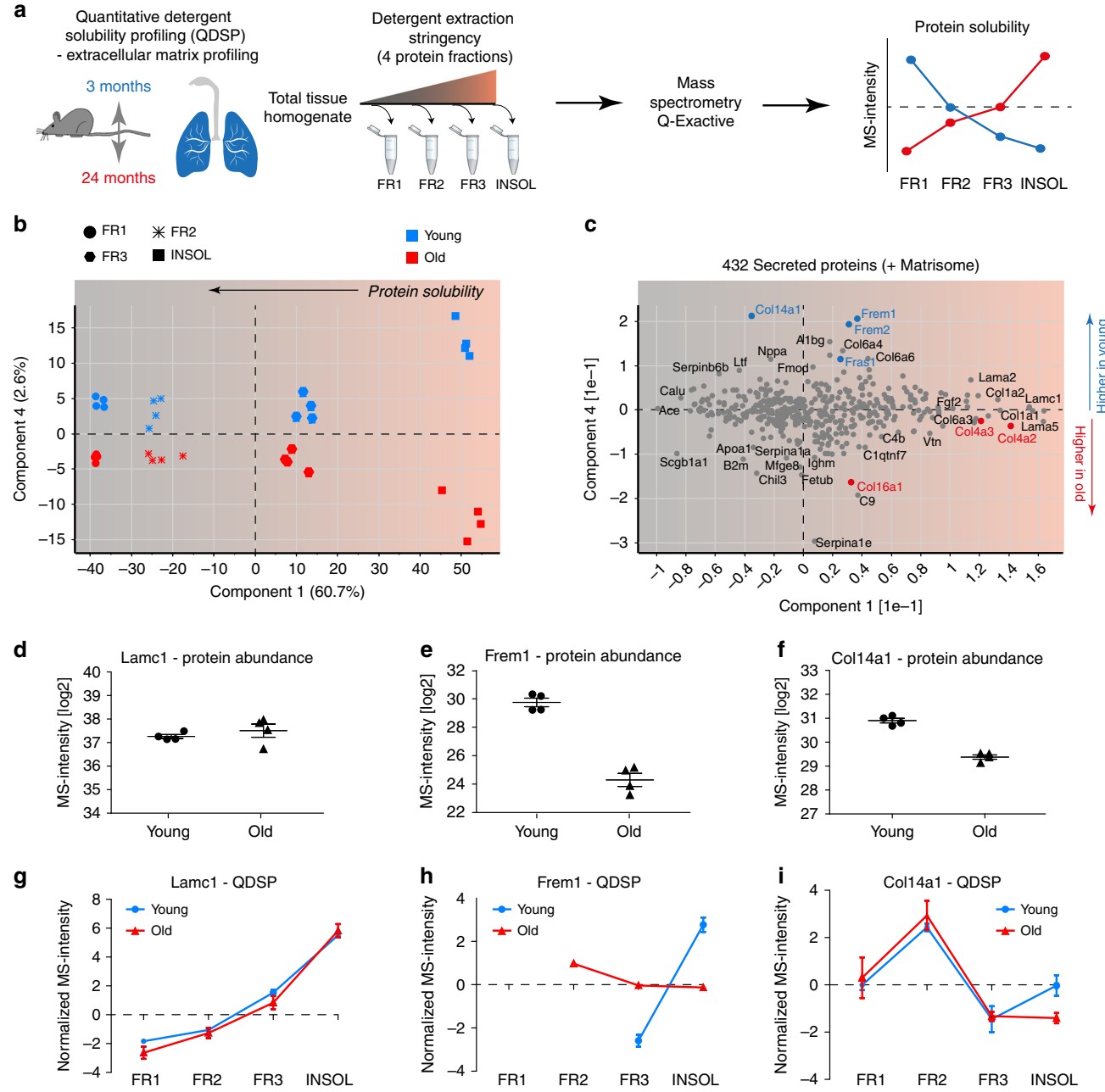

**Fig. 6** Proteome-wide detergent solubility profiling reveals changes in the extracellular matrix (ECM) architecture. **a** Experimental design—extraction of proteins from whole lung homogenates with increasing detergent stringency results in four distinct protein fractions, which are analyzed by mass spectrometry (MS). **b** The projections of a principal component analysis (PCA) of 432 proteins with the annotation 'secreted' in the Uniprot and/or Matrisome database separate the four protein fractions, indicated by symbol shape, in component 1 and the age groups, as indicated by color, in component 4. **c** The loadings of the PCA are shown. **d–f** Relative differences in MS intensity (abundance) of the indicated proteins. **g–i** The normalized MS intensity across the four protein fractions from differential detergent extraction highlights changes in protein solubility between young and old mice for the indicated proteins. Error bars represent the standard error of the means (n = 4)

remodeling[42]. Interestingly, we observed very similar gene expression changes in type-2 pneumocytes of old mice as reported for the *Insig1/2* deletion. Consistently, the upstream regulator analysis predicted increased activity of *Srebf2* and reduced activity of *Insig1* specifically in type-2 pneumocytes of old mice (Supplementary Fig. 6c). The upstream regulator analysis was based on 25 known targets of SREBP/Insig1, all of which were increased in aged type-2 pneumocytes (Fig. 8a). Using gene annotation enrichment analysis on the universal protein resource (Uniprot) Keywords, Gene Ontology (GO) terms, and Kyoto

encyclopedia of genes and genomes (KEGG) pathways (Supplementary Data 6), we found increased cholesterol biosynthesis as the top hit in type-2 pneumocytes and lipofibroblasts and no other cell type (Fig. 8b). Indeed, most of the *Insig1/2* target genes are directly involved in cholesterol biosynthesis (Fig. 8c).

To confirm the increased cholesterol biosynthesis and analyze the actual lipid content of the cells, we performed immunofluorescence of the type-2 pneumocyte marker prosurfactant protein C (proSP-C) together with the LipidTox compound that stains neutral lipids. Increased LipidTox staining in aged lungs

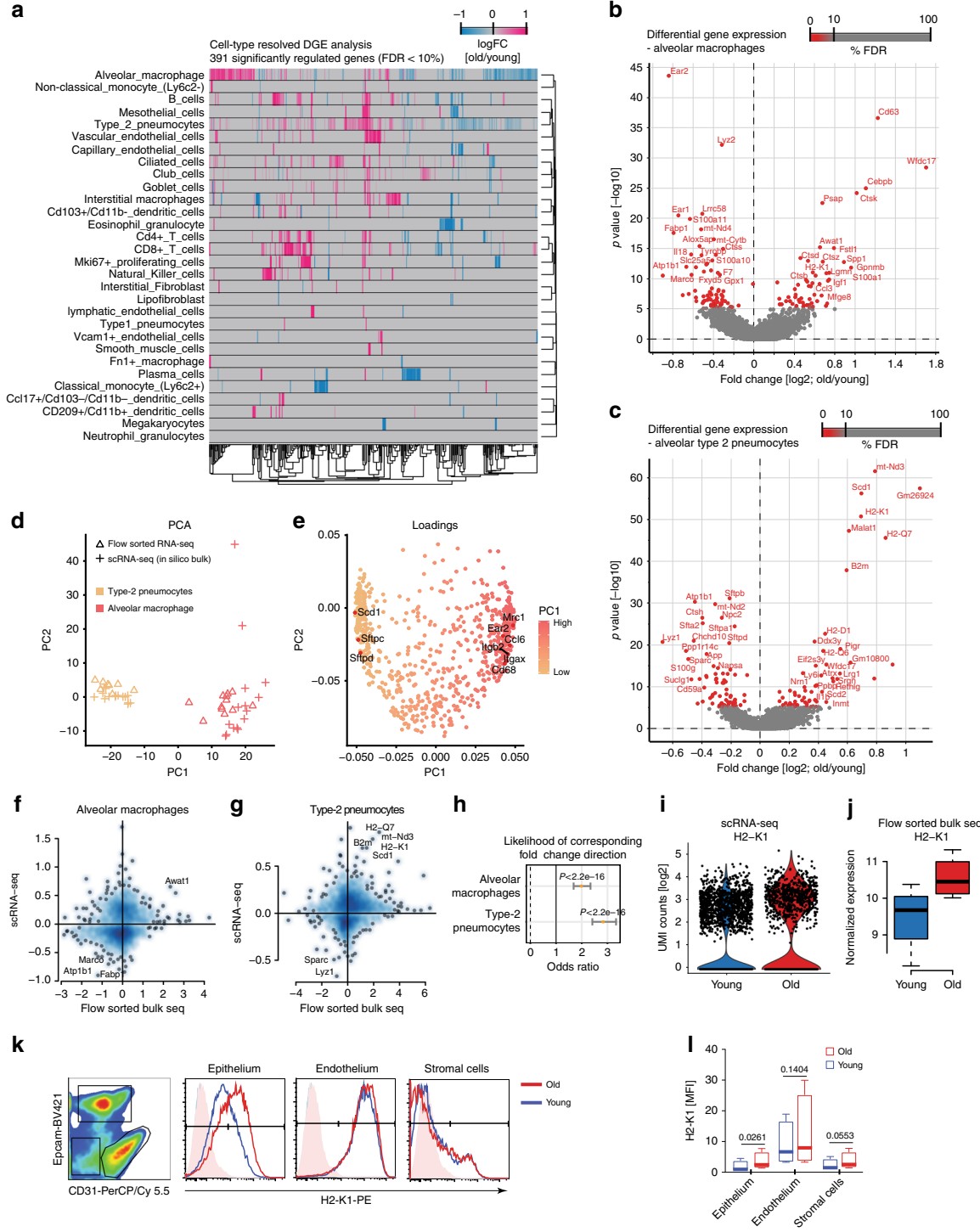

was specific to alveolar type-2 cells (Fig. 8d). In addition, we used the Nile red dye to stain neutral lipids in cells of a whole lung suspension after depletion of leukocytes. Using flow cytometry we quantified the Nile red lipid staining and found a significant increase in mean fluorescence intensity (Fig. 8e–g) in the CD45-negative cells of old mice. CD45+ cells were not significantly altered, indicating that the increase in neutral lipid content is specific to epithelial cells and fibroblasts. Thus, we have shown that increased cholesterol biosynthesis and neutral lipid content in type-2 pneumocytes and lipofibroblasts is a hallmark of lung aging.

## Discussion

Enabling healthy aging is one of the prime goals of the modern society. In order to better understand age-related chronic lung diseases such as COPD, lung cancer, or fibrosis, intense efforts in integrated multi-omics systems biology tools for the analysis of lung aging are needed[26]. In this work, we present a single-cell survey of mouse lung aging and computationally integrate single-cell transcriptomics data with bulk proteomics and transcriptomics of whole lung to build a draft of an atlas of the aging lung. Atlasing efforts are generally organized in stages so that more detailed maps of cellular phenotypes will be integrated at

**Fig. 7** Single-cell RNA-sequencing (scRNA-seq) enables cell type-resolved differential expression analysis. **a** Heatmap displays fold changes derived from the cell type-resolved differential expression analysis. Rows and columns correspond to cell types and genes, respectively. Negative fold change values (blue) represent higher expression in young compared to old. Positive fold change values are colored in pink. **b**, **c** Volcano plots visualize the differential gene expression results in **b** alveolar macrophages and **c** type-2 pneumocytes. X and Y axes show average log2 fold change and −log10 p value, respectively. **d** Scatterplot illustrates principal component analysis (PCA) of in silico bulk samples of alveolar macrophages and type-2 pneumocytes and the projected flow-sorted bulk samples. Color and shape indicate cell-type identity and data modality. PCA loadings show that well-known marker genes define the first principal component corresponding to cell-type identity (**e**). Fold changes derived from the flow-sorted bulk samples and the cell type-resolved differential expression analysis are depicted on the X and Y axes respectively for alveolar macrophages (**f**) and type-2 pneumocytes (**g**). The likelihood of corresponding fold change direction was highly enriched between the scRNA-seq and flow-sorted bulk data for both cell types (**h**). X-axis shows the odds ratio including 95% confidence interval. Black vertical line illustrates an odd ratio of one representing equal likelihood. Increased expression of H2-K1 in old compared to young mice was observed for type-2 pneumocytes in the scRNA-seq (**i**) and flow-sorted bulk (**j**) data (n = 4 young and n = 4 old mice). For (**j**), the box represents the interquartile range, the horizontal line in the box is the median, and the whiskers represent 1.5 times the interquartile range. **k** The indicated cell lineages were gated by flow cytometry as shown in the left panel in a CD31 and Epcam co-staining and evaluated for H2-K1 expression on protein level. The histograms show fluorescence intensity distribution of the H2-K1 cell surface staining for the indicated lineages and age groups. **l** Boxplot shows mean fluorescence intensity for H2-K1 in the indicated cell types taken from 4 young and 4 old mice. The p values are from a two-sided t-test. The box represents the interquartile range, the horizontal line in the box is the median, and the whiskers represent 1.5 times the interquartile range

later stages to initial drafts of the atlas. The intention in this study was to perform an integrated analysis of aging effects at a depth of current state of the art of proteomics and transcriptomics. The lung aging atlas and associated raw data can be accessed at https://theislab.github.io/LungAgingAtlas (Supplementary Fig. 7). It features five dimensions that can be navigated through gene and cell type-specific queries: (1) cell type-specific expression of genes and marker signatures for 30 cell types, (2) regulation of gene expression by age on cell-type level, (3) cell type-resolved pathway and gene category enrichment analysis, (4) regulation of protein abundance by age on tissue level, and (5) regulation of protein solubility by age.

The highly multiplexed nature of droplet-based single-cell RNA sequencing used in this study allows the direct analysis of thousands of individual cells freshly isolated from whole mouse lungs, providing unbiased classification of cell types and cellular states. Two previous studies have analyzed aging effects using single-cell transcriptomics and found increased transcriptional variability between cells in human pancreas and T cells[21,22]. In this study, we identify aging-associated increased transcriptional noise, which may result from deregulated epigenetic control, in most cell types of the lung, indicating that this phenomenon is a general hallmark of aging that likely affects most cell types in both mice and humans. This concept is supported by our study and it will be interesting when and how future investigations will shed light on the molecular mechanisms driving this phenomenon.

We have used three independent cohorts of young and old mice and uncovered remarkably well-conserved aging signatures in both mRNA and protein. Thus, the three datasets validate each other and show that (1) single-cell analysis can be highly representative of biological changes in total tissue, and (2) the analysis of protein and mRNA content can lead to overall similar results with important differences. Hallmarks of aging, such as the downregulation of mitochondrial oxidative phosphorylation and the upregulation of pro-inflammatory signaling pathways, were consistently observed in all datasets. On the level of individual genes/proteins, however, we often observed interesting differences, which indicates that for functional analysis of a particular gene/protein, it remains essential to also analyze the protein, which ultimately executes biological functions.

The example of basement membrane collagen IV genes that were all downregulated on the mRNA level but upregulated on the protein level illustrates that protein post-transcriptional regulation is indeed important. In particular, the abundance of ECM proteins, which often have long half-lives and are thus likely more often regulated on the posttranscriptional level, could frequently

show decoupling of protein and mRNA. Next to mass spectrometry-based methods, single-cell methods combining mRNA and protein analysis, such as cellular indexing of transcriptomes and epitopes by sequencing (CITE-seq)[43], will become ever more important in the near future. We show that the combination of single cell-resolved mRNA analysis and bulk proteomics is highly complementary using the single-cell expression data to understand the most likely cellular origin of proteins that showed altered abundance with age. Spatial transcriptomics methods for high-throughput detection of transcripts in single cells in situ are currently quickly evolving[44,45]. Traditional antibody-based methods for single-cell protein analysis in situ are however not well multiplexed and do not easily scale for high throughput. Thus, to fully develop the enormous potential of single-cell multi-omics data integration, the field depends on current and future developments in multiple omics layers on single-cell level in situ[46,47].

We analyzed the foundations of lung tissue architecture by quantifying compositional and structural changes in the aged extracellular matrix using state-of-the-art mass spectrometry workflows. The ECM is not only key as a scaffold for the lungs overall architecture, but also an important instructive niche for cell fate and phenotype[25,48]. Recent proteomic studies identified at least 150 different ECM proteins, glycosaminoglycans, and modifying enzymes in the lung, and these assemble into intricate composite biomaterials that are characterized by specific biophysical and biochemical properties[32,33,49]. Due to this complexity of the ECM, both in terms of composition and posttranslational modification and the assembly of ECM proteins into supramolecular structures, it is presently unclear on which level and how exactly the aging process affects the lung ECM scaffold. We used detergent solubility profiling to screen for differences in protein crosslinking and complex formation within the ECM. Surprisingly, most solubility profiles were not significantly altered with age, indicating that aging-related ECM remodeling does not involve large differences in covalent protein crosslinks. However, we observed a few very strong changes in the ECM which have not yet been reported in the context of aging and are open for future investigation into their functional implications.

In order to stabilize the alveolar structure during breathing-induced expansion and contraction, type-2 pneumocytes produce and secrete pulmonary surfactant, which is a thin film of phospholipids and surfactant proteins[41]. The lipid composition of pulmonary surfactant has been shown to change with age[50], and electron microscopy of surfactant and the lipid-loaded lamellar

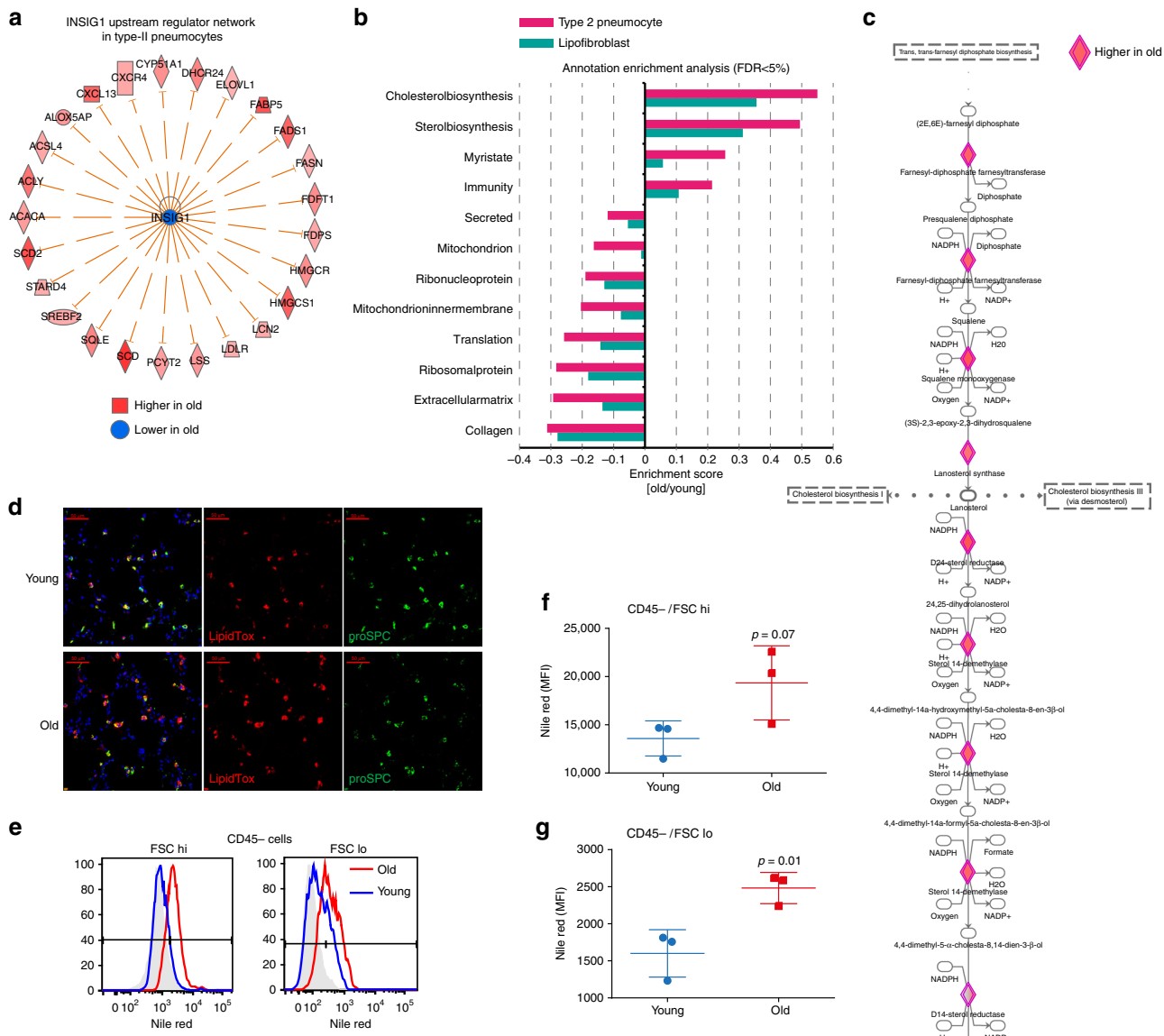

**Fig. 8** Aging increases cholesterol biosynthesis in type-2 pneumocytes and lipofibroblasts. **a** The graph shows genes known to be negatively regulated by Insig1 that were found to be upregulated in type-2 pneumocytes of old mice. **b** Selected gene categories found to be significantly (false discovery rate (FDR) < 5%) upregulated (positive enrichment scores) or downregulated (negative enrichment scores) in the indicated cell types. **c** Segment of the cholesterol biosynthesis pathway. Diamond-shaped nodes represent enzymes that were found to be upregulated in type-2 pneumocytes of old mice. The biochemical intermediates are named in between the enzyme nodes. **d** Immunofluorescence staining of lung sections of young and old mice shows type-2 pneumocytes expressing pro-SPC and neutral lipids marked by LipidTox staining (scale bar: 50 μm). **e** Quantification of Nile red stainings using flow cytometry. Histograms show flow cytometry analysis of Nile red in aged (red) and young (blue) mice; unstained control is represented in gray. Cells were stratified by size in bins of large (FSC hi) and small (FSC lo) cells using the forward scatter. **f**, **g** Nile red mean fluorescence intensity (MFI) quantification across three individual mice for **f** CD45-negative and forward scatter (FSC) high, and **g** FSC low cells. The *p* values are from an unpaired, two-sided *t*-test

bodies in type-2 pneumocytes revealed ultrastructural disorganization with age[51]. This may be related to our finding that cholesterol biosynthesis and neutral lipid content is upregulated in type-2 cells of old mice. It is currently unclear at which level the homeostasis of lipid metabolism is altered in the aged lungs. We found strong similarity of the aged type-2 phenotype with the phenotype in *Insig1/2* knockout mice that accumulated neutral lipids, accompanied by lipotoxicity-related lung inflammation and tissue remodeling[42]. Thus, it is possible that part of the chronic inflammation we observed in the aged lung is influenced by deregulation of lipid homeostasis. The inflammatory phenotype may also be related to epithelial senescence, as mice with a type-2 pneumocyte-specific deletion of telomerase, and thus

premature aging with increased senescence in these cells, developed a pro-inflammatory tissue microenvironment and were less efficient in resolving acute lung injury[52].

In summary, we have demonstrated that the lung aging atlas presented here contains a plethora of information on molecular and cellular scale and serves as a reference for the large community of scientists studying chronic lung diseases and the aging process.

## Methods

**Ethics statement**. Pathogen-free C57BL/6 mice were obtained from Charles River and housed in rooms maintained at constant temperature and humidity with a 12 h light cycle. Animals were allowed food and water ad libitum. For this study, organs were obtained from mice that had to be killed because of excessive breeding.

Animal handling was performed according to strict governmental and international guidelines and ethical oversight by the local government for the administrative region of Upper Bavaria, Germany.

**Generation of single-cell suspensions from whole mouse lung**. After killing mice, lung tissue was perfused with sterile saline from the right to the left ventricle of the heart and subsequently inflated via a catheter in the trachea by an enzyme mix containing dispase (50 caseinolytic units/ml), collagenase (2 mg/ml), elastase (1 mg/ml), and DNase (30 µg/ml). After tying off the trachea, the lung was removed and immediately minced to small pieces (approximately 1 mm$^2$). The tissue was transferred into 4 ml enzyme mix for enzymatic digestion for 30 min at 37 °C. Enzyme activity was inhibited by adding 5 ml of phosphate-buffered saline (PBS) supplemented with 10% fetal calf serum (FCS). Dissociated cells in suspension were passed through a 70 µm strainer and centrifuged at $500 \times g$ for 5 min at 4 °C. Red blood cell lysis (Thermo Fisher 00-4333-57) was done for 2 min and stopped with 10% FCS in PBS. After another centrifugation for 5 min at $500 \times g$ (4 °C) the cells were counted using a Neubauer chamber and critically assessed for single-cell separation and viability. A total of 250,000 cells were aliquoted in 2.5 ml of PBS supplemented with 0.04% of bovine serum albumin and loaded for DropSeq at a final concentration of 100 cells/µl.

**Single-cell RNA sequencing**. Dropseq experiments were performed according to the original Dropseq protocol[15,16]. Using a microfluidic polydimethylsiloxane device (Nanoshift), single cells (100/µl) from the lung cell suspension were co-encapsulated in droplets with barcoded beads (120/µl, purchased from ChemGenes Corporation, Wilmington, MA) at rates of 4000 µl/h. Droplet emulsions were collected for 15 min/each prior to droplet breakage by perfluorooctanol (Sigma-Aldrich). After breakage, beads were harvested and the hybridized mRNA transcripts reverse transcribed (Maxima RT, Thermo Fisher). Unused primers were removed by the addition of exonuclease I (New England Biolabs), following which beads were washed, counted, and aliquoted for pre-amplification (2000 beads/reaction, equals ~100 cells/reaction) with 12 PCR cycles (Smart PCR primer: AAGCAGTGGTATCAACGCAGAGT (100 µM), 2× KAPA HiFi Hotstart Readymix (KAPA Biosystems), cycle conditions: 3 min 95 °C, 4 cycles of 20 s 98 °C, 45 s 65 °C, 3 min 72 °C, followed by 8 cycles of 20 s 98 °C, 20 s 67 °C, 3 min 72 °C, then 5 min at 72 °C)[15]. PCR products of each sample were pooled and purified twice by 0.6× clean-up beads (CleanNA), following the manufacturer's instructions. Prior to tagmentation, complementary DNA (cDNA) samples were loaded on a DNA High Sensitivity Chip on the 2100 Bioanalyzer (Agilent) to ensure transcript integrity, purity, and amount. For each sample, 1 ng of pre-amplified cDNA from an estimated 1000 cells was tagmented by Nextera XT (Illumina) with a custom P5 primer (Integrated DNA Technologies). Single-cell libraries were sequenced in a 100 bp paired-end run on the Illumina HiSeq4000 using 0.2 nM denatured sample and 5% PhiX spike-in. For priming of read 1, 0.5 µM Read1CustSeqB (primer sequence: GCCTGTCCGCGGAAGCAGTGGTATCAACGCAGAGTAC) was used.

**Bioinformatic processing of scRNA-seq reads**. The Dropseq core computational pipeline15 was used for processing next-generation sequencing reads of the scRNA-seq data. STAR (version 2.5.2a) was used for mapping[53]. Reads were aligned to the mm10 genome reference (provided by the Dropseq group via the Gene Expression Omnibus (GEO) accession code GSE63269). For cell filtering, we considered all barcodes with more than 200 genes detected within the top 1200 barcodes by total UMI counts. Samples muc3838, muc3839, muc3840, and muc3841 were sequenced at lower depth in which case we considered the top 800, 500, 500, and 500 barcodes by total UMI counts corresponding to the expected number of cells, respectively.

**Single-cell data analysis**. After constructing the single-cell gene expression count matrix, we used the R package Seurat[54] and custom scripts for analysis.

For unsupervised clustering and visualization, we first defined highly variable genes within each mouse sample separately following the Seurat standard approach. Next, genes appearing in >4 mouse samples in the set of highly variable genes were defined as a set of consensus highly variable genes. To minimize the effect of cell cycle on clustering we removed cell-cycle genes[55] from the set of consensus highly variable genes. All 14,813 cells passing quality control were merged into one count matrix and normalized and scaled using Seurat's NormalizeData() and ScaleData() functions, in which we regressed out the total UMI count. The reduced set of consensus highly variable genes was used as the feature set for independent component analysis using Seurat's RunICA() function. The first 30 independent components were used for tSNE visualization and Louvain clustering using the Seurat functions RunTSNE() and FindClusters(), respectively.

To quantitatively assess the clustering overlap across mouse samples, the Silhouette coefficient was calculated. The Silhouette coefficient was calculated between the Euclidean distance of the 50 independent components and the mouse sample indicator. The Silhouette coefficient ranges from −1 to 1 and values close to zero indicate random clustering with regard to the specified indicator.

The Seurat FindAllMarkers() function was used to identify cluster-specific marker genes. Based on manual annotation and with guidance of the enrichment analysis (see below), the 36 clusters were assigned to 30 cell-type identities. Using the annotation of cell-type identities, the FindAllMarkers() function was called to identify the final set of cell-type markers used throughout this analysis.

An important technical detail needed our attention and is briefly described here. As infrequently discussed in the community but not yet addressed, we also observed 'ambient mRNA' effects, which we believe are the consequence of free mRNA released from dying cells hybridizing with beads in droplets during the microfluidic capture of single cells in the Dropseq workflow. The ambient mRNAs are typically derived from highly abundant transcripts and this artifact is inherent to all droplet-based methods (including the commercially available 10× platform). Here, it can be exemplified by the *Scgb1a1* gene in Fig. 1c that is known to be highly specific for club and goblet cells but was nevertheless detected in almost 100% of the cells in our data. However, the UMI count levels were much higher in club and goblet cells (representing the real source of expression), indicating that the mRNA counts observed in all other clusters were of ambient mRNA background. To independently confirm this we therefore determined all genes that showed ambient mRNA background by analyzing the identity of genes on beads at the tail-end of the total UMI count distribution (on average 10 UMIs per barcode), representing empty beads that were never in contact with a real cell but nevertheless contain information from free-floating ambient mRNA. We identified 153 genes (Supplementary Data 7) with an 'ambient mRNA' effect and accounted for this effect in the cell type-resolved differential expression analysis (see below for details).

To aid the assignment of cell type to clusters derived from unsupervised clustering, we performed cell-type enrichment analysis. Cell-type gene signatures obtained from bulk-level gene expression were downloaded from the ImmGen and xCell resources. Each gene signature obtained from our clustering was statistically evaluated for overlap with gene signatures contained in these two resources. Mouse gene symbols were capitalized to map to human gene symbols. Overlap between gene signatures was evaluated using Fisher's exact test.

Cell-type marker signatures in our data (Supplementary Data 1) were compared to cell-type marker signatures in the MCA[13]. MatchScore[20] was used to quantify overlap between cell-type marker signatures derived from our study and the MCA. Marker genes with adjusted $p$ value < 0.1 and average log fold change >1 were considered.

Transcriptional noise in the gene expression profiles was quantified following previous work[22]. For each cell type with at least 10 old and young cells, we quantified transcriptional noise in the following manner. To account for differences in total UMI counts, all cells were downsampled so that all cells had equal number of total UMI counts. To account for differences in cell-type frequency, cell numbers were down-sampled so that equal numbers of young and old cells were used. Next, genes were divided into 10 equally sized bins based on mean expression and the top and bottom bins excluded. Within each bin, the 10% of genes with the lowest coefficient of variation were selected. Subsampled raw count data were reduced to this set of genes and square-root transformed. Next, the euclidean distance between each cell and the corresponding cell-type mean within each age group was calculated. This euclidean distance was used as one measure of transcriptional noise for each cell. Additionally, we average the euclidean distances for each mouse and calculated the transcriptional noise ratio between young and old mice. Alternatively, we calculated Spearman's correlation coefficients on the down-sampled expression matrices across all genes between all pairwise cell comparisons within each cell type and age group. To be consistent with the sign of the metric we used 1–Spearman correlation coefficient as the second measure of transcriptional noise. To statistically assess the association between transcriptional noise and age within each cell type, Wilcoxon's rank sum test was used. The $p$ values were subsequently corrected for multiple testing using the Bonferroni–Hochberg method as implemented in the R function p.adjust().

Cell-type frequencies were calculated based on the counts of cells annotated to each cell type for each mouse. Counts were transformed to proportions using the DR_data() function of the DirichletReg R package which causes the values to shrink away from extreme values of 0 and 1. Next, the mouse-wise euclidean distances were calculated based on these proportions using the dist() R function followed by multidimensional scaling using the isoMDS() R function. To statistically assess the association between age and the first coordinate derived from the multidimensional scaling, Wilcoxon test was applied. Relative changes in cell-type frequencies were calculated by subtracting the median cell-type proportion of the young mice from the cell-type proportions of the old mice.

Cell type-resolved differential expression analysis was performed using the Seurat differential gene expression testing framework. Within each cell type, cells were grouped by age and differential testing performed using the Seurat FindMarkers() function. By inspecting barcodes with a very low number of UMI counts, we identified 153 potential ambient mRNAs. However, these mRNAs could represent true housekeeper genes which are constitutively expressed in all cells. Therefore, we removed 41 mRNAs which showed no cell type-specific expression effect (log2 fold change < 1) in any of the cell types in the cell-type marker discovery analysis from this list. Next, to avoid differential testing of a gene in a cell type where expression levels are driven by the ambient effect, cell type-resolved differential expression testing of the remaining 112 ambient mRNAs was limited to

cell types in which the ambient mRNA showed moderate cell type-specific expression (adjusted $p$ value < 0.25).

The one-dimensional annotation enrichment analysis[24] was used for cell type-resolved pathway analysis. We used the freely available software package Perseus[56], as previously described[33]. To predict the activity of upstream transcriptional regulators and growth factors based on the observed gene expression changes, we used the Ingenuity® Pathway Analysis platform (IPA®, QIAGEN Redwood City, www.qiagen.com/ingenuity). The analysis uses a suite of algorithms and tools embedded in IPA for inferring and scoring regulator networks upstream of gene expression data based on a large-scale causal network derived from the Ingenuity Knowledge Base. The analytics tool Upstream Regulator Analysis[23] was used to compare the known effect (transcriptional activation or repression) of a transcriptional regulator on its target genes to the observed changes to assign an activation Z-score. Since it is a priori unknown which causal edges in the master network are applicable to the experimental context, the Upstream Regulator Analysis tool uses a statistical approach to determine and score those regulators whose network connections to dataset genes as well as associated regulation directions are unlikely to occur in a random model[23]. In particular, the tool defines an overlap $p$ value measuring enrichment of network-regulated genes in the dataset, as well as an activation Z-score which can be used to find likely regulating molecules based on a statistically significant pattern match of up- and down-regulation, and also to predict the activation state (either activated or inhibited) of a putative regulator. In our analysis we considered genes with an overlap $p$ value of >7 (log10) that had an activation Z-score > 2 as activated and those with an activation Z-score < −2 as inhibited.

**Proteomics and multi-omics data integration**. For proteome analysis ~100 mg of fresh frozen total tissue (wet weight) of mouse lungs was homogenized in 500 μl PBS (with protease inhibitor cocktail) using an Ultra-turrax homogenizer. After centrifugation the soluble proteins were collected and proteins were extracted from the insoluble pellet in three steps using buffers with increasing stringency using the QDSP protocol[33]. Lungs were perfused with PBS through the heart to remove blood. Then, ~100 mg of total lung tissue (wet weight) was homogenized in 500 μl PBS (with protease inhibitor cocktail and EDTA) using an Ultra-turrax homogenizer. After centrifugation the soluble proteins were collected and proteins were extracted from the insoluble pellet in three steps using buffers with increasing stringency (*buffer 1*: 150 mM NaCl, 50 mM Tris-HCl (pH 7.5), 5% Glycerol, 1% IGPAL-CA-630 (Sigma, #I8896), 1 mM MgCl2, 1× Protease inhibitors (+EDTA), 1% Benzonase (Merck, #70746-3), 1× Phosphatase inhibitors (Roche, #04906837001); *buffer 2*: 50 mM Tris-HCl (pH 7.5), 5% Glycerol, 150 mM NaCl +fresh protease inhibitor tablet (+EDTA), 1.0% IGEPAL® CA-630, 0.5% sodium deoxycholate, 0.1% SDS, 1% Benzonase (Merck, #70746-3); *buffer 3*: 50 mM Tris-HCl (pH 7.5), 5% Glycerol, 500 mM NaCl, protease inhibitor tablet (+EDTA), 1.0% IGEPAL® CA-630, 2% sodium deoxycholate, 1% SDS, 1% Benzonase (Merck, #70746-3)). Insoluble pellets were resuspended in detergent containing buffers and incubated for 20 min on ice (except for buffer 3, which was used at room temperature), followed by separation of soluble and insoluble material using centrifugation for 20 min at 16,000 × g. The PBS from the tissue homogenate and the NP40 soluble fraction (buffer 1) was pooled which, together with the two fractions derived from ionic detergent extraction (buffer 2 and 3), resulted in a total of three soluble fractions and one insoluble pellet that were subjected to liquid chromatography-tandem mass spectrometry (LC-MS/MS) analysis. Soluble proteins were precipitated with 80% acetone and subjected to solution digestion using a modified published protocol[57]. In brief, protein reduction (10 mM TCEP) and alkylation (50 mM CAA) were performed at once in 6 M guadinium hydrochloride (100 mM Tris-HCl pH 8) at 99 °C for 15 min. Subsequent protein digestion was done in two steps. The first digestion was done at 37 °C for 2 h with LysC (1:50 enzyme to protein ratio) in 10 mM Tris-Hcl (pH 8.5) containing 2 M guadinium hydrochloride (Gdm), 2.7 M Urea, and 3% acetonitrile. The second digestion step was done using fresh LysC (1:50 enzyme to protein ratio) and trypsin (1:20 enzyme to protein ratio) in 600 mM Gdm, 800 mM Urea, and 3% acetonitrile at 37 °C overnight. For the insoluble protein pellet, which is strongly enriched for insoluble ECM proteins, we optimized the in-solution digestion protocol with additional steps involving extensive mechanical disintegration and ultra-sonication aided digestion. The insoluble material was cooked, reduced, and alkylated in 6 M Gdm for 15 min and then subjected to 200 strokes in a micro-dounce device, which reduced the particle size of the insoluble protein meshwork. We then proceeded with the two-step digestion protocol described above, which was additionally aided by 15 min ultrasonication (Bioruptor, Diagenode) in the presence of the enzymes in both digestion steps. Peptides were purified using stage-tips containing a polystyrene-divinylbenzene copolymer modified with sulfonic acid groups (SDB-RPS) material (3 M, St. Paul, MN 55144-1000, USA) as previously described[57].

Mass spectrometry data were acquired on a Quadrupole/Orbitrap type Mass Spectrometer (Q-Exactive, Thermo Scientific) as previously described[33]. Approximately 2 μg of peptides were separated in a 4 h gradient on a 50 cm long (75 μm inner diameter) column packed in-house with ReproSil-Pur C18-AQ 1.9 μm resin (Dr. Maisch GmbH). Reverse-phase chromatography was performed with an EASY-nLC 1000 ultra-high pressure system (Thermo Fisher Scientific), which was coupled to a Q-Exactive Mass Spectrometer (Thermo Scientific). Peptides were loaded with buffer A (0.1% (v/v) formic acid) and eluted with a nonlinear 240 min

gradient of 5–60% buffer B (0.1% (v/v) formic acid, 80% (v/v) acetonitrile) at a flow rate of 250 nl/min. After each gradient, the column was washed with 95% buffer B and re-equilibrated with buffer A. Column temperature was kept at 50 °C by an in-house designed oven with a Peltier element[58] and operational parameters were monitored in real time by the SprayQc software[59]. MS data were acquired with a shotgun proteomics method, where in each cycle a full scan, providing an overview of the full complement of isotope patterns visible at that particular time point, is followed by up to 10 data-dependent MS/MS scans on the most abundant not yet sequenced isotopes (top10 method)[60]. Target value for the full scan MS spectra was $3 \times 10^6$ charges in the 300−1650 $m/z$ range with a maximum injection time of 20 ms and a resolution of 70,000 at $m/z$ 400. Isolation of precursors was performed with the quadrupole at window of 3 Th. Precursors were fragmented by higher-energy collisional dissociation with normalized collision energy of 25% (the appropriate energy is calculated using this percentage, and $m/z$ and charge state of the precursor). MS/MS scans were acquired at a resolution of 17,500 at $m/z$ 400 with an ion target value of $1 \times 10^5$, a maximum injection time of 120 ms, and fixed first mass of 100 Th. Repeat sequencing of peptides was minimized by excluding the selected peptide candidates for 40 s.

MS raw files were analyzed by the MaxQuant[61] (version 1.4.3.20) and peak lists were searched against the human Uniprot FASTA database (version Nov 2016), and a common contaminants database (247 entries) by the Andromeda search engine[62] as previously described[33]. As fixed modification cysteine carbamidomethylation and as variable modifications, hydroxylation of proline and methionine oxidation was used. False discovery rate was set to 0.01 for proteins and peptides (minimum length of seven amino acids) and was determined by searching a reverse database. Enzyme specificity was set as C-terminal to arginine and lysine, and a maximum of two missed cleavages were allowed in the database search. Peptide identification was performed with an allowed precursor mass deviation up to 4.5 ppm after time-dependent mass calibration and an allowed fragment mass deviation of 20 ppm. For label-free quantification in MaxQuant the minimum ratio count was set to two. For matching between runs, the retention time alignment window was set to 30 min and the match time window was 1 min. The mass spectrometry proteomics data have been deposited to the ProteomeXchange Consortium via the PRIDE [1] partner repository with the dataset identifier PXD012307.

QDSP analysis intensities were first normalized such that the mean log2 intensities of the young and the old samples are zero, respectively. Using the normalized intensities, a two-way ANOVA with the two-factor treatment (old/young) and solubility fraction (FR1, FR2, FR3, INSOL) and the corresponding interaction term was performed using the R function aov(). Proteins significant in the interaction term correspond to proteins for which the solubility profile changes between young and old mice. Therefore, the corresponding $p$ value was used for filtering the significantly changed profiles after FDR correction.

**scRNA-seq, bulk RNA-seq, and proteome integration**. In silico bulk samples were generated by summing UMI counts across all cells within one mouse sample. Differential gene expression analysis of in silico bulk samples was performed using the R package DESeq2 (v1.20.0)[63].To integrate scRNA-seq, bulk RNA-seq, and protein data, the following approach was used. Raw count data from the in silico bulk and whole lung tissue bulk were normalized using the voom() function of the limma R package[64]. Next, in silico bulk, whole lung tissue bulk, and protein data were merged on a set of genes present in all three data sets and quantile normalized. This merged and quantile normalized expression matrix was then subjected to PCA.

Some statistical and bioinformatics operations, such as normalization, pattern recognition, cross-omics comparisons, and multiple-hypothesis testing corrections, were performed with the Perseus software package[56]. The two-dimensional annotation enrichment test used to compare proteome and transcriptome is based on a two-dimensional generalization of the nonparametric two-sample test. The false discovery rate is stringently controlled by correcting for multiple hypothesis testing[24].

**Flow cytometry**. Isolated total lung cell suspensions were used to detect and quantify cell populations and activation by flow cytometry. We depleted red blood cells by positive selection of Ter199 cells, followed by CD45 bead separation (Miltenyi Biotec; Bergish Gladbach, 130-052-301). Next, we analyzed cells by fluorescence-activated cell sorting (FACS) cell suspensions before and after CD45 separation and stained cell suspensions with anti-mouse CD31 (Biolegend, 102419), EpCAM (Biolegend, 118225), and H2-K1 (Thermo Fisher Scientific, Waltham, 12-5998-81). Cells were stained in the dark at 4 °C for 20 min. CD45 lineage-negative cells were stained with Nile red (Santa Cruz Biotechnology, sc-203747) in a 1:1000 dilution for 10 min at 4 °C, as previously reported[62].Cells were sorted using the CD45-negative fraction of the cell isolate stained for anti-mouse CD31, and EpCAM antibodies. Epithelial cells were sorted as CD31− cells and EpCAM+ cells. For sorting macrophages we used the CD45-positive fraction and stained with anti-mouse CD11c (Biolegend, 117310), CD11b (Biolegend, 101216), MHC II (Biolegend, 107615), Siglec-F (552126, BD Pharmingen), and Ly6G (Biolegend, 127627) antibodies. For flow cytometry sorting, neutrophils were excluded by selection of Ly6G-negative cells. Macrophages were sorted as MHCII+, CD11c+, CD11b+ as previously described[65]. Data acquisition was performed in

a BD Fortessa flow cytometer (Becton Dickinson, Heidelberg, Germany). All stainings were performed per 300,000 cells in the following dilutions: CD31 (1:300), EpCAM (1:50), H2-K1 (1:50), CD11c (1:100), Siglec-F(1:20), CD11b (1:25), MHCII (1:50), and Ly6G (1:10).

Data were analyzed using the FlowJo software (TreeStart Inc., Ashland, OR, USA). Data were reported as absolute numbers (cells/μl), normalized by bead counts (BD Truecount TM Beads tubes; BD Biosciences, Heidelberg, Germany) (Supplementary Fig 5). For H2-K1 and Nile red, data were analyzed by mean fluorescence intensity (MFI). Negative thresholds for gating were set according to isotype-labeled and unstained controls.

**Bulk RNA-sequencing and analysis**. RNA was isolated from whole lung tissue using the Qiagen RNeasy® Mini Kit (#74104) according to the manufacturer's recommendations. The RNA isolate was thereafter enriched for poly-A templates and submitted for whole mRNA sequencing on the Illumina HiSeq4000.

Whole lung tissue bulk RNA next-generation sequencing reads were aligned to the mouse reference genome mm10 using STAR[53] (version 2.2.1). Read summarization was performed using the featureCounts[63] (version 1.5.0) tool. To statistically evaluate the agreement between the in silico bulk and true bulk RNA-seq data, Spearman's correlation coefficients were calculated on the gene expression profiles between all sample pairs and the averages of both modalities. Differential gene expression analysis of whole lung tissue bulk samples was performed using the R package DESeq2[66] (v1.20.0).

To identify potential age-dependent alterations in tissue composition, the whole lung tissue bulk RNA-seq were integrated with the scRNA-seq-derived cell-type signatures. Kolmogorov–Smirnov test was used to statistically evaluate the enrichment of cell-type marker genes in the fold changes derived from the differential expression analysis of the whole lung tissue bulk RNA-seq. The $p$ values were limited to the range from 1 to 1e−50.

Flow-sorted macrophages and epithelial cells were immediately lysed after sorting and cDNA synthesis was performed using the Smart-Seq® v4 Ultra® Low Input RNA Kit for Sequencing (TaKaRa, 634896). For each sample, 200 pg of pre-amplified cDNA from an estimated 2000 cells was tagmented by Nextera XT (Illumina) according to the manufacturer's protocol and submitted for sequencing on the Illumina HiSeq4000.

Flow-sorted bulk RNA next-generation sequencing reads were aligned to the mouse reference genome mm10 using STAR[53] (version 2.2.1). Read summarization was performed using the featureCounts[63] (version 1.5.0) tool. To increase comparability between bulk and single-cell RNA-seq data, a total of 30 in silico bulk samples were generated by summing the counts from all cells belonging to the alveolar macrophages and type-2 pneumocytes clusters for each mouse. Next, PCA was calculated for these in silico bulk samples using the alveolar macrophages and type-2 pneumocytes marker genes with adjusted $p$ value < 0.1 and fold change > 0 (Supplementary Data 1). Subsequently, the flow-sorted bulk RNA-seq samples were projected into this PCA space to show correspondence between the scRNA-seq-derived in silico bulk samples and the flow-sorted RNA-seq samples. Differential expression analysis of flow-sorted bulk RNA-seq samples was conducted using the R package limma[64]. To statistically evaluate the agreement between the age-dependent alterations measured in the scRNA-seq and flow-sorted bulk RNA-seq data, Fisher's exact test was used. Fisher's exact test assesses the likelihood of genes having the same fold change direction (up- or down-regulation in old compared to young).

**Proximity ligation in situ hybridization (PLISH)**. Samples were prepared and processed for PLISH and immunostaining as described in Nagendran et al.[67] with some modifications. 14 μm mouse lung cryosections were collected on superfrost slides and allowed to air dry for 10 min. The slides were immersed in prewarmed 10 mM citrate buffer containing 0.05% lithium dodecyl sulfate at 100 °C in a water bath for 5 min. The slides were quickly removed, rinsed with diethyl pyrocarbonate (DEPC)-treated water and air dried. Seal chambers (GBL621505 Sigma-Aldrich) were applied and the sections were rehydrated with DEPC-treated water for 1 min. The samples were incubated with 0.025 mg/ml Pepsin (10108057001 Roche; from Sigma-Aldrich) in 0.1 M HCL for 5 min at 37 °C followed by a quick rinse with 1× PBS and the addition of H probes for Col4a1.

H probe sequences were: Col4a1 NM_009931.2:mmHLC2-VB01-Col4a1-5315 AGGTCAGGAATACTTACGTCGTTATGGTAGGGTTCATTGCTGTTACA, mmHRC2-VB01-Col4a1-5315 AGGTCACACAGGATATAATTCTTATAGGTCGAGTAGTATAGCCAGGTT, mmHLC2-VB01-Col4a1-5385 AGGTCAGGAATACTTACGTCGTTATGGAGTTACGCGAATCCCTATAA, mmHRC2-VB01-Col4a1-5385 CCAACGAAGCGGGGTGTGTTTTATAGGTCGAGTAGTATAGCCAGGTT, mmHLC2-VB01-Col4a1-5910 AGGTCAGGAATACTTACGTCGTTATGGTTGACCTGCCTAATTCTGA, mmHRC2-VB01-Col4a1-5910 AACAGGCTCTACGCTAGAACTTATAGGTCGAGTAGTATAGCCAGGTT, mmHLC2-VB01-Col4a1-5848 AGGTCAGGAATACTTACGTCGTTATGGATTTATTTATTTTCCATCTA, mmHRC2-VB01-Col4a1-5848 ATATATATATTTATTTACTTTTATAGGTCGAGTAGTATAGCCAGGTT,

mmHLC2-VB01-Col4a1-5753 AGGTCAGGAATACTTACGTCGTTATGGAAGTTTGTGTTTGGGGCTGA, mmHRC2-VB01-Col4a1-5753 CATAGTACCACACAGGGCATTTATAGGTCGAGTAGTATAGCCAGGTT.

Connector circle CCC2.1: 5′ ATTCCTGACCTAACAAACATGCGTCTATAGTGGAGCCACATAAT-TAAACCTGGCTAT 3′.

Variable bridge VB01-P1: ACTACTCGACCTATAACCATAACGACGTAAGT.

Label probe: LP1m-Cy5: 5′Cy5/ CTATACTACTCGACCTATA.

**Immunofluorescence and histology**. For immunofluorescence microscopy, mouse lungs were perfused with PBS, fixed in 4% paraformaldehyde (pH 7.0), and embedded in paraffin for formalin-fixed, paraffin-embedded sections. The paraffin sections (3.5 μm) were deparaffinized and rehydrated, and the antigen retrieval was accomplished by pressure-cooking (30 s at 125 °C and 10 s at 90 °C) in citrate buffer (10 mM, pH 6.0). After blocking for 1 h at room temperature with 5% bovine serum albumin, the lung sections were incubated with the primary antibodies overnight at 4 °C, incubated with the secondary antibodies (1:250) for 2 h, followed by 4′,6-diamidino-2-phenylindole (DAPI; Sigma-Aldrich, 1:2000) for 20 min at room temperature. Images were acquired with an LSM 710 microscope (Zeiss). The following primary (1) and secondary (2) antibodies were used: (1) CC10 rabbit (Santa Cruz, sc-25554, 1:100), Foxj1 mouse (Santa Cruz, sc-53139, 1:50), collagen IV rabbit (Abcam, ab6586, 1:100), (2) donkey anti-mouse Alexa Fluor (AF) 647 (Invitrogen, A21447), donkey anti-rabbit AF 568 (Invitrogen, A10042), and donkey anti-goat AF 488 (Invitrogen, A21202). Counterstain with LipidTox was performed using HCS LipidTOX deep red neutral lipid stain (Invitrogen, H34477, 1:200).

The frequency of ciliated (nuclear Foxj1+) and club cells (CC10+) were quantified by counting 2647 cells, covering a total length of 22 mm airway in 28 individual airways (young, $n = 14$; old $n = 14$) of 2 mice of each age group. We normalized cell numbers to the total length of their respective airway using the ZEN 2.3 SP1 software for image processing.

**Reporting summary**. Further information on experimental design is available in the Nature Research Reporting Summary linked to this article.

**Code availability**. The code to reproduce the analyses and figures described in this study can be found at: github.com/theislab/2018_Angelidis.

## Data availability

Proteome raw data can be downloaded from the PRIDE repository under the accession number PXD012307. scRNA-seq, whole lung tissue bulk and flow-sorted cell populations bulk raw data, can be downloaded from the Gene Expression Omnibus under the accession number GSE124872. The whole lung aging atlas can be accessed via an interactive user-friendly webtool at: https://theislab.github.io/LungAgingAtlas. All other data supporting the findings of this study are available from the corresponding authors upon reasonable request.

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

## Acknowledgements

We thank Silvia Weidner and Daniela Dietel for excellent technical assistance. We also thank Gabi Sowa, Igor Paron and Korbinian Mayr for expert support of the proteomics pipeline. We thank Sandy Lösecke for technical assistance in next generation sequencing and Thomas Schwarzmayr for support with High-seq 4000 sequencing raw data. L.M.S. acknowledges funding from the European Union's Horizon 2020 research and innovation programme under the Marie Sklodowska-Curie grant agreement No 753039. This work was supported by the German Center for Lung Research (DZL), the Helmholtz Association, and the German Federal Ministry of Education and Research (BMBF), project Single Cell Genomics Network Germany.

## Author contributions

H.B.S. conceptualized and supervised the entire project and wrote the paper. F.J.T. supervised single-cell analysis and multi-omics data integration. I.A. and M.S. performed single-cell transcriptomics experiments. L.M.S., I.A., M.A., and H.B.S. analyzed single-cell transcriptomics data. H.B.S. performed proteomics experiments. H.B.S. and L.M.S. analyzed the proteomics data and performed transcriptomics and proteomics data integration. I.A. and C.H.M. performed histology and immuno-fluorescence microscopy. I.E.F. and F.R.G. performed flow cytometry experiments. M.N. and T.D. performed PLISH experiments. E.G. and T.-M.S. performed next-generation sequencing of single-cell libraries. L.M.S. and G.T. set up the interactive webtool. O.E. assisted in data interpretation and M.M. provided important support with mass spectrometry equipment. All authors read, edited, and approved the manuscript.

## Additional information

**Competing interests:** The authors declare no competing interests.

