## [Peer Review File · Nature Communications]

Reviewers' comments:

Reviewer #1 (Remarks to the Author):

This is an interesting and important resource paper that provides data describing normal aging in the mouse lung. Specifically, the investigators have combined in droplet based single cell RNA-Seq with unbiased tissue proteomics to describe aging in the normal lung. The data are of high quality and the analyses are carefully planned and executed. A useful web based tool is provided with the data that will enhance its utility for the research community. In addition, some biologic insights have been made from both the proteomic and the transcriptional data and validated using IHC or flow, providing examples of how these data can be used. I have some concerns that can be readily addressed.

1. The globally compare the proteomic data with the single cell RNA seq data, the investigators "summed" the transcriptomes from individual cell populations to create an "in silico bulk" RNA-Seq dataset. The rationale behind this analysis is unclear. Such "in silico bulk" is inherently affected by the biases introduced by the tissue composition and single cell isolation procedures and does not provide additional information about the age-associated changes within individual cellular populations. If this analysis was performed in order to assess the completeness of the "atlasing" effort, then it should be compared to an appropriate real bulk RNA-seq performed on whole lung tissue obtained from the same experimental animals. Similarly aged animals from the same colony should suffice to generate these data, which could be compared with the "in silico bulk" analysis (likely to show the limitations of single cell data) and to compare with the bulk proteome.

2. To assess the age-related changes in gene expression within individual cell types, the authors compared young and old animals using the FindAllMarkers function in the Seurat package. This somewhat undermines the power of single cell analysis as it treats samples similar to bulk RNA-seq. Specifically, the analysis makes an a priori assumption that all animals aged at the same rate and that aging did not introduce heterogeneity within the cellular populations within the same animal. It would be interesting if authors could re-analyze of at least some of the well-represented cell types (like alveolar type II cells or airway epithelial cells) by subsetting them and subjecting them to an additional round of re-clustering. This analysis would provide important information regarding the age-associated changes and likely would lend support the data regarding the increased transcriptional noise with aging.

3. In Figure S3, the investigators undertook an analysis to suggest a reduced ratio of club cells relative to ciliated cells in the aging airways. Quantification of cell populations based on single cell RNA-Seq data is difficult given limitations to tissue digestion protocols. This is particularly germane

to this analysis, as the authors clearly demonstrate age related changes in the extracellular matrix that might alter the susceptibility to cellular liberation. The concern is somewhat offset by the confirmatory IHC, however, this in itself is problematic as a systematic approach (e.g. stereology) was not undertaken for careful quantification. I would recommend either eliminating this analysis or adding substantial caveats in the discussion of these data.

4. The investigators make note of a populations of cells enriched for cell cycle genes (Fig 1b, cluster 2). These have been frequently reported in single cell data but their origins and importance have yet to be validated. It would be interesting to regress out the cell cycle genes and see if these cells all end up in a single cluster or multiple clusters (i.e. have marker genes characteristic of a single, or multiple cell populations). It would also be interesting to see if these cells change with aging.

5. The investigators mention they identified 153 genes they attributed to ambient RNA based on their identification on beads unlikely to have been in contact with any cells. They state they accounted for these genes in the analysis but this methodology is not completely described or referenced. This methodology would be important for the field.

6. As the conclusion that aging increases transcriptional noise is a major finding of the paper, it would be important to have a better understanding of the statistical methods that were undertaken. Specifically, was transcriptional noise increased only when all cells were considered as a composite, or were there increases in transcriptional noise within particular cell populations during aging. In considering this, it is important to treat each mouse as an individual "n" (i.e. n=7 and n=8) as the chance for false discovery seems high if each cell is used as a separate statistical event. Even if the latter was performed, it is not clear if the differences were significant (after correcting for multiple comparisons) between any individual cell population or if the data were significant only in aggregate. If only the aggregate (ANOVA) is significant, I am not sure one can say the transcriptional noise "increased" or just "changed" as the reduction in in transcriptional noise in plasma cells was quite large relative to other populations and likely drove many of the changes in the ANOVA.

7. Figure 4 shows upregulation of H2-K1 in epithelial cells of old mice (with a smaller upregulation in other cell populations), and Figure 6 shows upregulation of several genes involved in cholesterol biosynthesis. These novel biologic insights are nicely validated with flow cytometry. Were these changes (or the changes in Scd1) detected in the proteomic data. If not, a discussion of possible reasons would be informative. Minor concern, the young and aged data in Figure 4g should be reversed (young before old) for readability.

8. Figure 5 is difficult to understand, both from the description in the text and from the figure. The analysis seems to be highly speculative in terms of its conclusions about the upstream regulators driving changes in gene expression, and seems to distract from the validated biological insights presented in Fig 4 and Fig. 6. I wonder if these data might be moved to the supplement. If they are to stay in the main paper, a more detailed discussion of the methodology and the strengths/limitations of the analysis should be included.

9. The investigators show relatively poor correlation between single cell RNA seq data and proteome data for some proteins (particularly those in the extracellular matrix, Fig 2e). It is at least worth discussing the observation that many matrix proteins in the lung have extremely long half lives (months). Therefore, these data should not be necessarily interpreted as a disconnect between transcriptome and proteome but instead emphasize the need for serial measures over the lifespan.

10. Related to many of the comments above, it would be important to include the code used for the data analysis as part of the paper or as a linked resource, understanding that methods for analyzing these data will likely evolve over time.

11. I was confused by Figure 3J/K. --how were the mesenchymal cells identified were cells from the individual mice annotated and clustered or was this a sub-analysis of the combined object? Is Figure 3K highlighting marker genes for these clusters or does it have something to do with aging. Not sure these two panels add much to the overall figure.

Reviewer #2 (Remarks to the Author):

Angelidis et al. present a resource in which single-cell transcriptomes and the bulk proteome of the lungs of young and old mice are compared. Some aspects are of general interest and go beyond a comparative analysis of lung aging. In particular, the authors discuss that their dataset allows them dissect sources of bulk proteome and transcriptome changes during aging with cell type resolution.

Unfortunately, the work is mostly descriptive and does not contain in-depth validations and insights into underlying molecular mechanisms, except a few stainings that partially confirm expression patterns derived from scRNASeq/proteomics. Providing such more detailed validations for selected 1-2 cases would demonstrate the usefulness of their dataset to the community and could greatly increase the interest to a broader readership.

In its present form, the paper remains vague and represents a general discussion of cherry-picked examples from diverse aspects of lung physiology and aging. I therefore believe that it would be suited for a more specialized journal, unless in-depth validations are provided in a revision. Below, I discuss the strengths and weaknesses of the article in more detail using the same chronology as in the article.

1) Single-cell transcriptomes of ~15,000 cells are used to compare gene expression in young and old mice. The authors conclude that: (a) there is a good overlap between samples; (b) cell type identity was not strongly confounded by aging and (c) transcriptional noise increases during aging. These statements should be supported by more quantitative measures: In Figure S1B it appears that the cell type identity actually changes quite substantially during the aging process, and in Fig S1A the overlap is hard to judge. Transcriptional noise may also mainly reflect technical variation, changes in the cell type frequencies with age and systematic differential expression in each cell type (the latter two phenomena occur as the authors state further below). Can the authors exclude a major contribution of such confounding factors?

2) The dataset is further enriched using bulk proteomics obtained from an independent cohort of mice. The authors show that the differentially expressed genes during aging are enriched for similar biological functions and upstream regulatory pathways when compared at the level of proteomics and (in silico reconstructed) bulk RNASeq. While such a conceptual similarity is encouraging and interesting, the authors should also provide the concrete overlap between transcriptome and proteome changes. The opposing changes of mRNA and protein for Collagen IV should also be validated at the RNA level, ideally using a spatially resolved setup such as RNA FISH, to prove that the combined proteome-transcriptome analysis indeed provided novel insights into post-transcriptional regulation.

3) The authors state that their dataset can be used to predict the cellular source of regulated proteins in the proteomics from scRNASeq data, while distinguishing real changes in gene expression from alterations in cell type frequencies. I find this a very intriguing application, but the description remains vague and the authors mainly refer to an online tool, without performing any validation using follow-up experiments (see also comment 5 below).

4) The following section indicates that the solubility of proteins does not globally change during aging. The authors identify a few potentially interesting solubility changes with aging (not validated independently). This section a bit disconnected from the rest of the work, needs better motivation and could be better shortened.

5) The following sections on cell-type specific effects during aging reveal potentially interesting insights into how gene expression is selectively changed in certain cell-types. The results in this section show that ~120 genes are differentially expressed in each of the two most abundant cell types with age, thus questioning the authors' previous statement that cell type identity is preserved during aging. This needs clarification.

The subsequent validations focus on type 2 pneumocytes. For one MHC class protein, cell-type specific differential expression was confirmed at the protein level using flow cytometry (Fig. 4). Furthermore, Nile Red staining, a proxy for cellular cholesterol levels, is shown to be increased in old mice as predicted based on increased RNA expression in the cholesterol synthesis pathway (Fig. 6). The authors should better explain the nature of the markers used in Figs. 4F, 6D and E, and should clarify how specific they are for alveolar type 2 pneumocytes. Fig. 6: Why were cells stratified there using the forward scatter?

The current validations are restricted to one specific example protein (MHC) or the use of an indirect marker (Nile Red). Furthermore, the physiological relevance of these findings is unclear. To show that their resource is indeed useful to the community, the authors must provide a global validation of their predictions for a larger number of genes. I suggest that they use the existing (or a similar) flow cytometry setup (with sorting) followed by global analysis (RNA sequencing and possibly proteomics) to confirm that they can reliably predict cell-type-specific gene expression changes (also in relation comment 3).

Reviewer #3 (Remarks to the Author):

The manuscript presented by Angelidis et al combines single cell transcriptomics and 'deep' proteomics approaches associated with a panel of bioinformatics and biostatistical tools in an attempt to create a predictive reference map of cellular source of regulated proteins associated with the aging of the lung in a mouse model. It is an exhaustive investigation supported by an exemplary, reproducible and statistically credible experimental design, which results were validated by orthogonal technologies. The respective raw data are available online, as well as the whole lung aging atlas. It is therefore an important contribution not only to the understanding of the lung proteome itself but mainly to the events associated with the aging process in an animal model.

In summary, I strongly recommend the publication of this Angelidis et al 's manuscript in the Nat Comm.

Minors

Abbreviations should be written completely as much as possible when first mentioned in the text.

- Introduction session:

ECM (pag1) . If written in full in the Introduction session then it must be abbreviated in the 'results session'(pag4).

- Results session:

Dropseq (pag2)

tSNE map (pag2)

scRNAseq (pag 6)

Uniprot (pag 8)

GO (pag 8)

KEGG (pag 8)

- Discussion session

CITE-seg (pag 10)

- Methods session

PDMS (pag 17)

UMI (pag 17)

There are some 'typing errors' such as () in the Methods session that need to be fixed, for example:
pg 18: Seurat's `NomalizedData()`; `scaleData()`, et, etc....

Figure 6C: Please increase the letter size of the cholesterol biosynthesis pathway to make it readable.

Reviewer #4 (Remarks to the Author):

Angelidis and colleagues present an atlas of lung cells from aging mice. Moreover, they provide a link of single cell data to bulk proteomics data with excellent correlation between RNA and protein data strongly supporting the enormous value of single cell RNA-seq data. Furthermore, as an atlas resource the Theis lab provides a web interface for these data and also link the data to other atlas efforts such as the mouse cell atlas. As a resource paper, it is expected to stay mainly descriptive, albeit some attempts have been made to validate some of the work at least phenotypically. With the enormous expertise of the group of authors and laboratories, it is not surprising that the manuscript is very well presented. Nevertheless, since all atlas efforts are critically shaping the landscape of how cells are described during the next decades, several points are to be addressed first.

1. Albeit ~15,000 cells are a significant number of cells, it is still on the low side for a full organ in the mouse. Particularly, since this is the number of cells for both young and old animals. Moreover, the two major cell types, alveolar macrophages and type 2 pneumocytes make up a substantial portion of the overall number of cells, which leads to very few events for rare cells in the dataset. Furthermore, the authors correctly cite a paper from 2008 stating already 40 cell types, yet the authors only discover 30 cell types. Three things need to be added to the manuscript. One, a comparison of the 30 cell types with the 40 previous cell types and a reason why the other once might not have been detected in this new study. In other words, why are we falling behind current knowledge about existing cell types. Second, additional cells should be added which specifically might add to getting closer to the previously described 40 cell types. Third, a general remark in the discussion section, how such atlases are going to be build up further in the future by e.g. adding additional cells later to the dataset. The reader needs to understand the dynamics of such atlases, the current limitations (cell number, sequencing depth, any technical biases) and how this will be solved in the future.

2. Material and Methods Section: The authors describe red blood cell lysis as a step of generating single cell suspensions. How did the authors quantify contamination with red blood cells, and therefore with blood? How do the authors exclude that the remaining cells do not have any blood-derived contaminations? What about the immune cell compartment (e.g. T cells, B cells, monocytes). Is there a comparison with blood derived immune cells to see differences between the tissue-resident immune cells and blood-derived immune cells? These are important technical issues, particularly for the lung and require further explanation. Without any statement about quantification of the potential contamination of blood-derived cells, the lung atlas might be a mixed lung-blood atlas. The reader and user of the atlas needs this information.

3. The age-related changes of the major cell types is a major asset of the manuscript. In alveolar macrophages, Marco as an altered cell surface molecule is highlighted. Similar to the alveolar type 2 pneumocytes (flow cytometry for MHC) should be added for Marco on alveolar macrophages.

4. Figure 1: It is not entirely clear from the main result text, how the authors named the cells in Fig 1d (it is written in the methods section, but could be presented more prominent, in the end this is what a cell atlas does). For some cells (e.g. dendritic cells), it would be good to use updated nomenclature or at least add this nomenclature. What about subtypes of known cell types? For example, some of the immune cells identified are known to exist in subtypes. Explain, why you did not add this level of information. Or alternatively, point out that this is available on the website.

The clusters in Fig. 1 B cannot be linked to the cell nomenclature used in Fig. 1c and d. This is very difficult for the reader to follow. What are Fn1+ macrophages in context of the existing literature?

5. Figure 2: The RNA-seq and protein data are strikingly correlated, except for Collagen IV. Is there any functional or molecular explanation for the different behavior, while all other systems/pathways were correlated? If you do not have any proof, can you hypothesize?

6. Figure 2 and S3: Age dependent alterations of relative frequencies: How did the authors ensure that there was no technical reason for differences in relative frequencies? Did the authors control their relative frequencies with another independent single cell methodology? If not, such information should be added.

7. Figure 3: "Indeed, we were able to confirm the mesothelium specific expression of Fras1 using our scRNAseq data". How is this statement supported by the visualization in Fig. 3j/k? Fig. 3l is mentioned in the text, but does not exist. The information for Decorin is not included in Fig. 3j/k. In general, the authors need to make sure that figures, gene names and genes/proteins mentioned in text better match.

8. How many cells came from old, how many cells from young mice? One can infer from the text and the supplement, but it is not explicit. Add in main result text.

9. First sentence in discussion not necessary, it is a political statement. Increasing health, probably everybody would agree. Increasing lifespan is a very much debated statement. Enabling healthy aging might be ok again. Re-phrase.

10. Page 9: sentence of main text: This is a statement that is too negative and should be avoided. In This paper should stay away from the old discussion whether RNA could be used instead of protein. As long as proteomics cannot provide single cell data to a similar extend as scRNA-seq is doing it at the moment, these statements are not helpful.

11. Page 10, first paragraph: citation still in doi-format

12. Bioinformatic processing of scRNA-seq reads: Please provide some more statistics concerning the quality of the data. How many barcodes were detected, how many made the chosen cutoff, how many UMIs and genes per cell were recovered, mean read length, etc. Provide as a supplementary figure.

13. Data availability paragraph requires addition of the respective numbers instead of XXX.

Revision of NCOMMS-18-19103-T

Title: 'An atlas of the aging lung mapped by single cell transcriptomics and deep tissue proteomics'

Point by point reply to the reviewers

Reviewer #1 (Remarks to the Author):

This is an interesting and important resource paper that provides data describing normal aging in the mouse lung. Specifically, the investigators have combined in droplet based single cell RNA-Seq with unbiased tissue proteomics to describe aging in the normal lung. The data are of high quality and the analyses are carefully planned and executed. A useful web based tool is provided with the data that will enhance its utility for the research community. In addition, some biologic insights have been made from both the proteomic and the transcriptional data and validated using IHC or flow, providing examples of how these data can be used. I have some concerns that can be readily addressed.

We would like to thank the reviewer for taking the time to evaluate our work. We are grateful for the very constructive criticism and addressed your concerns as outlined below.

1. *The globally compare the proteomic data with the single cell RNA seq data, the investigators "summed" the transcriptomes from individual cell populations to create an "in silico bulk" RNA-Seq dataset. The rationale behind this analysis is unclear. Such "in silico bulk" is inherently affected by the biases introduced by the tissue composition and single cell isolation procedures and does not provide additional information about the age-associated changes within individual cellular populations. If this analysis was performed in order to assess the completeness of the "atlasing" effort, then it should be compared to an appropriate real bulk RNA-seq performed on whole lung tissue obtained from the same experimental animals. Similarly aged animals from the same colony should suffice to generate these data, which could be compared with the "in silico bulk" analysis (likely to show the limitations of single cell data) and to compare with the bulk proteome.*

We agree with the reviewer that the correct comparison here is a true bulk transcriptome from whole tissue. Thus, we now performed bulk RNA-seq from 3 young (3 month) and 3 old (22 months) mouse whole lung samples to repeat the analysis. The new data is presented in revised Figure 3 and the new Figure S4. With this new data we are now assessing age dependent changes in 3 independent cohorts of young and old mice with three complementary methods (scRNAseq, RNAseq, and mass spectrometry). As suggested by the reviewer we use the whole lung bulk RNAseq to assess the quality of the single cell suspension that was used for scRNAseq. Even though a potential bias with cell isolation could be expected we were very pleased to observe strong agreement between the real and in silico bulk data (see revised Fig. 3b), thus excluding strong biases by the single cell isolation procedures. Furthermore, we also observed strong correspondence of the age-dependent alterations in all three data sets (in silico bulk, real bulk and protein data - see PCA

analysis in revised Figure 3c). We have edited the Results and Methods section to describe these novel analyses.

“Figure 3. Multi-omic data integration uncovers common and distinct features in RNA and protein regulation. ... (b) On the left, gene expression profiles from whole lung bulk samples (n = 6) and in silico bulk samples (n = 15) were averaged and plotted on X and Y axes, respectively. Red and black lines indicate linear model fit and the diagonal. On the right, correlation heatmap shows the Pearson correlation between all bulk and in silico bulk samples. (c) Normalized bulk (RNA-seq) and in silico bulk (scRNA-seq) data was merged with proteome data (Mass spectrometry) and quantile normalized. The first two principal components show clustering by data modality. The third principal component separates young from old samples across all three data modalities. Blue and red colors indicate young and old samples, respectively.”

2. To assess the age-related changes in gene expression within individual cell types, the authors compared young and old animals using the *FindAllMarkers* function in the *Seurat* package. This somewhat undermines the power of single cell analysis as it treats samples similar to bulk RNA-seq. Specifically, the analysis makes an a priori assumption that all animals aged at the same rate and that aging did not introduce heterogeneity within the cellular populations within the same animal. It would be interesting if authors could re-analyze of at least some of the well-represented cell types (like alveolar type II cells or airway epithelial cells) by subsetting them and subjecting them to an additional round of re-clustering. This analysis would provide important information regarding the age-associated changes and likely would lend support the data regarding the increased transcriptional noise with aging.

As the reviewer correctly points out the Wilcoxon rank sum test implemented in the FindAllMarkers function does not implicitly model heterogeneity within the cellular populations within the same animal. However, with sufficient sample size (number of cells) non-parametric tests perform equally well compared to more elaborate modeling approaches and can robustly identify changes in single cell RNA-seq gene expression analysis (Pubmed ID: 29481549). To demonstrate this, we followed the reviewer’s suggestion and re-analyzed the airway epithelial cells. Expectedly, the three epithelial cell types formed three clusters (Fig. 4c). Both young and old cells cover the majority of the data manifold (Fig. 4d), indicating that there exists overlap in the gene expression profiles of young and old cells. However, we observed significant differences in the density of young and old cells across the data manifold.

“Figure 4. Whole lung cell type deconvolution reveals increase of Ciliated cells in airways of old mice. ... (c) The Fruchterman-Reingold (FR) embedding of the airway epithelial cells in the dataset reveals distinct clusters of airway cell identity. (d) The indicated color code shows the distribution of young and old cells to the three clusters presented in panel (c). ...”

The reviewer makes a good suggestion that, in theory, re-clustering of the cell types could visualize our finding regarding the increased transcriptional noise with aging. However, interpretation of distances in non-linear dimension reduced space, such as tSNE, is non-trivial as scaling is not consistent across the plot. Therefore, to additionally support our finding that transcriptional noise increases with aging, we used a second approach to quantify transcriptional noise. In the new approach we used Spearman correlation to quantify transcriptional noise. The results confirmed our previous finding (Fig. 2c and d).

“Figure 2. Most cell types show increased transcriptional noise with aging. (c) Scatterplot depicts the \log_2 ratio of transcriptional noise between old and young samples as calculated using 1 - Spearman correlation and the Euclidean distance between cells on the X and Y axes, respectively. For both panels, the size of the dots corresponds to the negative \log_{10} adjusted p-value of the cell type resolved differential transcriptional noise test and the red lines corresponds to the robust linear model regression fit. (d) As an example, the distribution of 1 - Spearman correlation coefficients between all pairs of young and old cells is shown for Type 2 pneumocytes. Larger values represent increased transcriptional noise. Blue and red colors indicate young and old samples.”

3. *In Figure S3, the investigators undertook an analysis to suggest a reduced ratio of club cells relative to ciliated cells in the aging airways. Quantification of cell populations based on single cell RNA-Seq data is difficult given limitations to tissue digestion protocols. This is particularly germane to this analysis, as the authors clearly demonstrate age related changes in the extracellular matrix that might alter the susceptibility to cellular liberation. The concern is somewhat offset by the confirmatory IHC, however, this in itself is problematic as a systematic approach (e.g. stereology) was not undertaken for careful quantification. I would recommend either eliminating this analysis or adding substantial caveats in the discussion of these data.*

We agree with the concern about limitations of relative frequency observations using single cell analysis of cells released from tissues. We therefore added the sentence ‘Relative frequency differences in scRNA-seq data can be biased by tissue isolation artefacts’ to the results section. Bulk tissue RNA-seq enabled us to additionally validate the change in ciliated cell proportions in whole lung by deconvolving the bulk expression data using our single cell gene expression profiles (see revised Figure 4 below). Indeed, we found that the ciliated cell marker genes signature was significantly upregulated in whole tissue bulk RNA-seq of old lungs (Fig. 4e/f in revised manuscript). For *in situ* validation we counted more than 2000 cells from a large number of airways and several mice (Fig. 4g-i). Thus, we did not want to remove this data and hope that the reviewer agrees that evidence from three independent experiments showing similar results would justify to show this data.

“Figure 4. Whole lung cell type deconvolution reveals increase of Ciliated cells in airways of old mice. (a) The MDS plot shows the mouse-wise euclidean distances of cell type proportions for the two age groups (b) The box plot shows the significant difference in the multidimensional scaling component 1 of cell type proportions between young and old. (c) The Fruchterman-Reingold (FR) embedding of the airway epithelial cells in the dataset reveals distinct clusters of airway cell identity. (d) The indicated color code shows the distribution of young and old cells to the three clusters presented in panel (c). Note the increased density of old cells in the Ciliated cell cluster. (e) The volcano plot shows negative log₁₀ enrichment p-values of cell type marker signatures in the differential expression results of the bulk RNA-seq data from young and old mice. (f) The empirical density plot shows significant enrichment for Ciliated cell type marker genes (red line) compared to all other genes (black line) in the distribution of fold changes derived from the bulk differential expression analysis. (g) Club and Ciliated cells were stained using a CC10 and Foxj1 antibody respectively. (h) The boxplot depicts the quantification of Ciliated cells from counting a total of 2647 Club and Ciliated cells in 14 individual airways of two mice of each indicated age group. (i) Ratio of Ciliated to Club cells in 14 individual airways of two mice for each indicated age group. P-values are derived from an unpaired, two-tailed t-test using Welch’s correction.”

4. The investigators make note of a populations of cells enriched for cell cycle genes (Fig 1b, cluster 2). These have been frequently reported in single cell data but their origins and importance have yet to be validated. It would be interesting to regress out the cell cycle genes and see if these cells all end up in a single cluster or multiple clusters (i.e. have marker genes characteristic of a single, or multiple cell populations). It would also be interesting to see if these cells change with aging.

We thank the reviewer for a constructive suggestion. Indeed, we observed one cluster of mainly proliferating cells. Following the reviewer's suggestion we re-analyzed this cluster in more detail and added the analysis into the Results section:

“Additionally, we noticed one cluster of mainly proliferating cells showing high expression levels for S and G2M cell cycle marker genes (Fig. S3a and b). Young mice showed a higher fraction of cells in this cluster compared to old mice (Fig. S3c; Generalized linear binomial model, $P < 0.001$). Next, we isolated this cluster and corrected the gene expression levels for cell cycle phase (Fig. S3 d and e). Subsequent unsupervised clustering analysis revealed that these proliferating cells belong to T cells, type 2 pneumocytes and alveolar macrophages (Fig. S3 f - i).”

The results from this analysis are described in an additional supplementary figure (Fig. S3):

Figure S3

“Supplementary Figure S3. Cell-cycle analysis reveals reduced proliferative capacity of T cells, Alveolar macrophages and Type-2 pneumocytes in aged lungs. (a, b) The ‘Mki67+ proliferating cell’ cluster (Fig. 1) showed high expression of (a) G2M- and (b) S-phase cell cycle signatures. (c) A higher fraction of proliferating cells was observed in young compared to old mice. (d) PCA based on cell cycle marker genes revealed clustering by cell cycle phase and (e) the removal of this effect after regressing out the cell cycle effect. Cells are colored by cell cycle phase as assigned by Seurat. (f) Unsupervised Louvain clustering revealed three distinct cell clusters. (g-i) tSNE visualization colored by the expression of cell type marker genes (g) Trbc2, (h) Sftpd and (i) Ear2 corresponding to T cells, Type 2 pneumocytes and alveolar macrophages, respectively.”

5. The investigators mention they identified 153 genes they attributed to ambient RNA based on their identification on beads unlikely to have been in contact with any cells. They state they accounted for

these genes in the analysis but this methodology is not completely described or referenced. This methodology would be important for the field.

We agree with the reviewer that we need to be more descriptive. Therefore, we point the reader in the “ambient RNA identification” section towards the “cell type resolved differential expression analysis” section where we describe our methodology in more detail.

“We identified 153 genes (Table S7) with an ‘ambient mRNA’ effect and accounted for this effect in the cell type resolved differential expression analysis (see below for details).”

Here, we extended the cell type resolved differential expression analysis section in the Methods section:

“Cell type resolved differential expression analysis. Cell type resolved differential expression analysis was performed using the Seurat differential gene expression testing framework. Within each cell type cells were grouped by age and differential testing performed using the Seurat FindMarkers() function. By inspecting barcodes with a very low number of UMI counts, we identified 153 potential ambient mRNAs. However, these mRNAs could represent true housekeeper genes which are constitutively expressed in all cells. Therefore, we removed 41 mRNAs which showed no cell type specific expression effect (\log_2 foldchange < 1) in any of the cell types in the cell type marker discovery analysis from this list. Next, to avoid differential testing of a gene in a cell type where expression levels are driven by the ambient effect, cell type resolved differential expression testing of the remaining 112 ambient mRNAs was limited to cell types in which the ambient mRNA showed moderate cell type specific expression (adjusted p-value < 0.25).”

6. As the conclusion that aging increases transcriptional noise is a major finding of the paper, it would be important to have a better understanding of the statistical methods that were undertaken. Specifically, was transcriptional noise increased only when all cells were considered as a composite, or were there increases in transcriptional noise within particular cell populations during aging. In considering this, it is important to treat each mouse as an individual "n" (i.e. n=7 and n=8) as the chance for false discovery seems high if each cell is used as a separate statistical event. Even if the latter was performed, it is not clear if the differences were significant (after correcting for multiple comparisons) between any individual cell population or if the data were significant only in aggregate. If only the aggregate (ANOVA) is significant, I am not sure one can say the transcriptional noise “increased” or just “changed” as the reduction in in transcriptional noise in plasma cells was quite large relative to other populations and likely drove many of the changes in the ANOVA.

Given the feedback from multiple reviewers we adapted our transcriptional noise analysis. We removed the global ANOVA result and instead focus on testing for a change in transcriptional noise within each cell type (Fig. 2a).

To further substantiate our results we quantified transcriptional noise in two manners: 1) Euclidean distance of the expression profile between cells and the cell type mean and 2) 1 - Spearman correlation coefficient calculated between all pairs of cells within one age group. In the current analyses we specifically account for differences in total UMI counts, cell type frequencies by down-sampling UMI counts and cells.

With respect to the reviewer's concerns regarding the sample size in the statistical analysis we additionally averaged the transcriptional noise metrics for each mouse and then calculated the ratio between old and young. These ratios calculated on the cell level (n= number of cells per cell type) and mouse averages (n=15) correlate significantly (Fig. 2b). Furthermore, the results of both the Euclidean distance and Spearman correlation based approaches validate each other (Fig. 2c). One notable exception are the Plasma cells which show opposite patterns in the Euclidean distance and Spearman correlation based approaches. However, the change in transcriptional noise between young and old mice for the Plasma cells was not statistically significant (Fig. 2c, as indicated by the size of the dot).

We have adapted the Results section accordingly:

"Therefore, we quantified transcriptional noise following previous work²¹ and accounted for differences in total UMI counts and cell type frequencies (see Methods for details). We observed an increase in transcriptional noise with aging in most cell types (Fig. 2a). To further exclude technical confounding we additionally averaged the transcriptional noise scores per mouse and obtained highly concordant results (Fig. 2b). To further substantiate this finding we quantified transcriptional noise in an alternative manner using Spearman correlations between cells. This analysis confirmed our finding that transcriptional noise is increased with aging (Fig. 2 c and d) and are in line with previous reports in the human pancreas²¹ or mouse CD4+ T cells²⁰."

Figure 2. Increased transcriptional noise in most cell types of old mice. (a) Boxplot illustrates transcriptional noise by age and celltype. Blue and red colors indicates young and old cells, respectively. Asterix indicates significant changes (Adjusted p-value <0.05). (b) Scatterplot shows the log2 ratio of transcriptional noise between old and young samples as calculated using mouse averages (n = 15) and single cells (n = number of cells) on the X and Y axes, respectively. (c) Scatterplot depicts the log2 ratio of transcriptional noise between old and young samples as calculated using 1 - Spearman correlation and the Euclidean distance between cells on the X and Y axes, respectively. For both panels, the size of the dots corresponds to the negative log10 adjusted p-value of the cell type resolved differential transcriptional noise test and the red lines corresponds to the robust linear model regression fit. (d) As an example, the distribution of 1 - Spearman correlation coefficients calculated between all pairs of young and old cells is shown for Type 2 pneumocytes. Larger values represent increased transcriptional noise. Blue and red colors indicate young and old samples.

7. Figure 4 shows upregulation of H2-K1 in epithelial cells of old mice (with a smaller upregulation in other cell populations), and Figure 6 shows upregulation of several genes involved in cholesterol biosynthesis. These novel biologic insights are nicely validated with flow cytometry. Were these changes (or the changes in *Scd1*) detected in the proteomic data. If not, a discussion of possible reasons would be informative. Minor concern, the young and aged data in Figure 4g should be reversed (young before old) for readability.

Current sensitivity of mass spectrometry does not enable detection of all genes we describe to be altered using the transcriptomic analysis. With a depth of >6000 proteins quantified in total lung tissue we did not detect *Scd1* in the proteome analysis and therefore cannot make a statement on protein regulation of this gene. Given the fact that indeed lipid content was found to be changed we argue that at least some of the enzymes in cholesterol biosynthesis must have been upregulated also on protein level.

We reversed the order of boxplots in Figure 7l accordingly.

8. Figure 5 is difficult to understand, both from the description in the text and from the figure. The analysis seems to be highly speculative in terms of its conclusions about the upstream regulators driving changes in gene expression, and seems to distract from the validated biological insights presented in Fig 4 and Fig. 6. I wonder if these data might be moved to the supplement. If they are to stay in the main paper, a more detailed discussion of the methodology and the strengths/limitations of the analysis should be included.

We moved this analysis to the supplement (Figure S6 in the revised manuscript).

9. The investigators show relatively poor correlation between single cell RNA seq data and proteome data for some proteins (particularly those in the extracellular matrix, Fig 2e). It is at least worth discussing the observation that many matrix proteins in the lung have extremely long half lives (months). Therefore, these data should not be necessarily interpreted as a disconnect between transcriptome and proteome but instead emphasize the need for serial measures over the lifespan.

We agree that poor correlation between mRNA and protein may be relevant in particular for ECM proteins, because the half life of ECM proteins can be very long and may therefore be regulated more often by postranscriptional mechanisms such as proteolytic degradation.

We added the following sentence to the discussion:

‘In particular the abundance of ECM proteins, which often have long half lives and are thus likely more often regulated on the posttranscriptional level could frequently show decoupling of protein and mRNA.’

10. Related to many of the comments above, it would be important to include the code used for the data analysis as part of the paper or as a linked resource, understanding that methods for analyzing these data will likely evolve over time.

We have made all analysis code available for reproducibility on github and point this out in the Data availability statement:

“Code to reproduce the analysis and figures described in this manuscript can be found at: github.com/theislab/2018_Angelidis.”

11. I was confused by Figure 3J/K. --how were the mesenchymal cells identified were cells from the individual mice annotated and clustered or was this a sub-analysis of the combined object? Is Figure 3K highlighting marker genes for these clusters or does it have something to do with aging. Not sure these two panels add much to the overall figure.

This was a subanalysis of the combined object. The only point for these panels was to illustrate the power of scRNAseq to identify cellular sources of proteins with the example of Col14a1 expressed by interstitial fibroblasts. Since the additional sub-clustering of mesenchymal cells did not really strengthen this aspect we removed that analysis and instead show now expression specificity of Col14a1 and its binding partner Decorin in the main data object (described in Figure 1). The new panel can be found in revised Figure 5.

“Figure 5. scRNA-seq predicts cellular origin of age-dependent protein alterations. ... (c) The dot plot shows mRNA expression specificity of Col14a1 and its binding partner Decorin (Dcn) in the scRNA-seq data.”

Reviewer #2 (Remarks to the Author):

Angelidis et al. present a resource in which single-cell transcriptomes and the bulk proteome of the lungs of young and old mice are compared. Some aspects are of general interest and go beyond a comparative analysis of lung aging. In particular, the authors discuss that their dataset allows them dissect sources of bulk proteome and transcriptome changes during aging with cell type resolution.

Unfortunately, the work is mostly descriptive and does not contain in-depth validations and insights into underlying molecular mechanisms, except a few stainings that partially confirm expression patterns derived from scRNASeq/proteomics. Providing such more detailed validations for selected 1-2 cases would demonstrate the usefulness of their dataset to the community and could greatly increase the interest to a broader readership.

In its present form, the paper remains vague and represents a general discussion of cherry-picked examples from diverse aspects of lung physiology and aging. I therefore believe that it would be suited for a more specialized journal, unless in-depth validations are provided in a revision. Below, I discuss the strengths and weaknesses of the article in more detail using the same chronology as in the article.

We would like to thank the reviewer for taking the time to evaluate our work and are grateful for the very constructive criticism. In particular, we agree with the reviewer that additional validations are necessary, therefore we specifically generated new RNA-seq data from two additional cohorts of mice. In the following we addressed your concerns as outlined below:

1) Single-cell transcriptomes of ~15,000 cells are used to compare gene expression in young and old mice. The authors conclude that: (a) there is a good overlap between samples; (b) cell type identity was not strongly confounded by aging and (c) transcriptional noise increases during aging. These statements should be supported by more quantitative measures: In Figure S1B it appears that the cell type identity actually changes quite substantially during the aging process, and in Fig S1A the overlap is hard to judge. Transcriptional noise may also mainly reflect technical variation, changes in the cell type frequencies with age and systematic differential expression in each cell type (the latter two phenomena occur as the authors state further below). Can the authors exclude a major contribution of such confounding factors?

To quantitatively assess the overlap between samples, we used the Silhouette coefficient. The Silhouette coefficient calculated between the Euclidean distance matrix of the 30 independent components and the mouse labels was close to zero (-0.074), indicating that the (cell type) clustering was random with respect to the mouse replicates and therefore no batch effect in the clustering was observed. To make this more clear we added the Silhouette coefficient into the main text:

“We observed very good overlap across mouse samples (Silhouette coefficient: -0.074) and most clusters were derived from >70% of the mice of both age groups (Fig. S1d and e).”

Additionally we added the following text into the Methods section:

“Quantitative assessment of overlap. To quantitatively assess the clustering overlap across mouse samples the Silhouette coefficient was calculated. The Silhouette coefficient was calculated between the Euclidean distance of the 30 independent components and the mouse sample indicator. The Silhouette coefficient ranges from -1 to 1 and values close to zero indicate random clustering with regards to the specified indicator.”

With respect to the reviewer’s comment regarding confounding cell type identity with aging, we would like to highlight that we took particular care in the definition of highly variable genes used for clustering and subsequent cell type identity discovery. We have prepared a detailed response in our reply to your point (5) (see below).

With respect to the reviewer’s comment regarding confounding differences in cell type frequencies or age-related expression changes with the results from our transcriptional noise analysis, we have improved our original transcriptional noise analysis. In particular, we quantified transcriptional noise in two manners: 1) Euclidean distance of the expression profile between cells and the cell type mean and 2) 1 - Spearman correlation coefficient calculated between all pairs of cells within one age group. In the current analyses we specifically account for differences in total UMI counts, cell type frequencies by down-sampling UMI counts and cells. We have edited the Results section to clarify this:

“Therefore, we quantified transcriptional noise following previous work²¹ and accounted for differences in total UMI counts and cell type frequencies (see Methods for details). We observed an increase in transcriptional noise with aging in most cell types (Fig. 2a). To further exclude technical confounding we additionally averaged the transcriptional noise scores per mouse and obtained highly concordant results (Fig. 2b). To further substantiate this finding we quantified transcriptional noise in an alternative manner using Spearman correlations between cells. This analysis confirmed our finding that transcriptional noise is increased with aging (Fig. 2 c and d) and are in line with previous reports in the human pancreas²¹ or mouse CD4+ T cells²⁰.”

Figure 2. Most cell types show increased transcriptional noise with aging. (a) Boxplot illustrates transcriptional noise by age and celltype. Blue and red colors indicates young and old cells, respectively. Asterix indicates significant changes (Adjusted p-value < 0.05). Cell types are ordered by decreasing transcriptional noise ratio between old and young cells. (b) Scatterplot shows the log2 ratio of transcriptional noise between old and young samples as calculated using mouse averages (n = 15) and single cells (n = number of cells) on the X and Y axes, respectively. (c) Scatterplot depicts the log2 ratio of transcriptional noise between old and young samples as calculated using 1 - Spearman correlation and the Euclidean distance between cells on the X and Y axes, respectively. For both panels, the size of the dots corresponds to the negative log10 adjusted p-value of the cell type resolved differential transcriptional noise test and the red lines corresponds to the robust linear model regression fit. (d) As an example, the distribution of 1 - Spearman correlation coefficients between all pairs of young and old cells is shown for Type 2 pneumocytes. Larger values represent increased transcriptional noise. Blue and red colors indicate young and old samples.

Additionally, we updated the Methods section to contain the new analysis:

“Quantifying transcriptional noise: Transcriptional noise in the gene expression profiles was quantified following previous work²¹. For each cell type with at least 10 old and young cells, we quantified transcriptional noise in the following manner. To account for differences in total UMI counts all cells were downsampled so that all cells had equal number of total UMI counts. To account for differences in cell type frequency, cell numbers were down-sampled so that equal numbers of young and old cells were used. Next, genes were divided into 10 equally sized bins based on mean expression and the top and bottom bins excluded. Within each bin the 10% of genes with the lowest coefficient of variation were selected. Subsampled raw count data was reduced to this set of genes and square-root transformed. Next, the euclidean distance between each cell and the corresponding cell type mean within each age group was calculated. This euclidean distance was used as one measure of transcriptional noise for each cell. Additionally, we average the euclidean distances for each mouse and calculated the transcriptional noise ratio between young and old mice. Alternatively, we calculated the Spearman correlation coefficients on the down-sampled expression matrices across all genes between all pairwise cell comparisons within each cell type and age group. To be consistent with the sign of the metric we used $1 - \text{Spearman correlation coefficient}$ as the second measure of transcriptional noise. To statistically assess the association between transcriptional noise and age within each cell type Wilcoxon’s rank sum test was used. P-values were subsequently corrected for multiple testing using the Bonferroni-Hochberg method as implemented in the R function `p.adjust()`.”

2) The dataset is further enriched using bulk proteomics obtained from an independent cohort of mice. The authors show that the differentially expressed genes during aging are enriched for similar biological functions and upstream regulatory pathways when compared at the level of proteomics and (in silico reconstructed) bulk RNASeq. While such a conceptual similarity is encouraging and interesting, the authors should also provide the concrete overlap between transcriptome and proteome changes. The opposing changes of mRNA and protein for Collagen IV should also be validated at the RNA level, ideally using a spatially resolved setup such as RNA FISH, to prove that the combined proteome-transcriptome analysis indeed provided novel insights into post-transcriptional regulation.

For the revised manuscript we performed bulk RNA-seq from 3 young (3 month) and 3 old (22 months) mouse whole lung samples to better establish the quality of the in silico bulk transcriptome from the scRNASeq data and additionally validate individual genes such as Collagen IV mRNA from whole tissue RNA-seq. The new data is presented in revised Figure 3 and the new Figure S4. With this new data we are now assessing age-dependent changes in 3 independent cohorts of young and old mice with three complementary methods (scRNAseq, RNAseq, and mass spectrometry) and consistently find Collagen IV mRNA reduction in both transcriptome datasets. Even though a potential bias with cell isolation could be expected we were very pleased to observe strong agreement between the real and in silico bulk data (see revised Fig. 3b), thus excluding strong biases by the single cell isolation procedures. Furthermore, we also observed strong correspondence of the

age-dependent alterations in all three data sets (in silico bulk, real bulk and protein data - see PCA analysis in revised Fig. 3c).

“We observed strong agreement between the real and in silico bulk data, thus excluding strong biases by the single cell isolation procedures (Fig. 3b). Furthermore, we also observed strong correspondence of the age-dependent alterations in all three datasets (Fig. 3c), indicating that we were able to pick up robust age dependent changes with three independent experimental settings. Significant correlation was observed between the gene-level fold changes derived from RNA-seq, scRNA-seq and protein expression data (Fig. S4d-f).”

“**Figure 3. Multi-omic data integration uncovers common and distinct features in RNA and protein regulation.** ... (b) On the left, gene expression profiles from whole lung bulk samples (n = 6) and in silico bulk samples (n = 15) were averaged and plotted on X and Y axes, respectively. Red and black lines indicate linear model fit and the diagonal. On the right, correlation heatmap shows the Pearson correlation between all bulk and in silico bulk samples. (c) Normalized bulk (RNA-seq) and in silico bulk (scRNA-seq) data was merged with proteome data (Mass spectrometry) and quantile normalized. The first two principal components show clustering by data modality. The third principal component separates young from old samples across all three data modalities. Blue and red colors indicate young and old samples, respectively. ...”

We have added the concrete, gene-level overlap between transcriptome and proteome changes into the Results section and refer to novel panels in Supplemental Figure S4:

“Supplementary Figure S4. Multi-omics lung aging data displays significant correspondence. Volcano plots show the significantly regulated genes from (a) in from scRNA-seq, (b) bulk RNA-seq and (c) mass spectrometry. (d-f) Differential expression results from multi-omics experiments show significant correspondence. X and Y axes illustrate the log2 fold changes calculated from the (d) RNA-seq and scRNA-seq (in silico bulk) experiments, (e) the mass spectrometry (protein) and scRNA-seq (in silico bulk) experiments, and (f) the mass spectrometry (protein) and RNA-seq experiments. Blue line indicates the Deming regression fit. Black dotted horizontal and vertical lines indicate 0 values (no differential expression) for the in silico bulk and protein data, respectively.”

As suggested by the reviewer we validated the Collagen IV finding using FISH. Using Proximity Ligation In Situ Hybridization (PLISH), we found reduced mRNA for Col4a1 in cryopreserved tissue sections of old mice compared to young mice. The new data is shown in revised Figure 2.

3) The authors state that their dataset can be used to predict the cellular source of regulated proteins in the proteomics from scRNASeq data, while distinguishing real changes in gene expression from alterations in cell type frequencies. I find this a very intriguing application, but the description remains vague and the authors mainly refer to an online tool, without performing any validation using follow-up experiments (see also comment 5 below).

To better validate our single cell differential expression analysis on a global scale we generated bulk RNA-seq from sorted cell populations. See our reply to point 5 below.

In revised Figure 5 we are showing Col14a1 as an example for prediction of the cellular source of a regulated protein (see panel below).

“Figure 5. scRNA-seq predicts cellular origin of age-dependent protein alterations. ... (c) The dot plot shows mRNA expression specificity of Col14a1 and its binding partner Decorin (Dcn) in the scRNA-seq data.”

We added the following sentence to the results section: ‘From the 5138 proteins quantified in the tissue proteome (Fig. 5a), we identified 32 Matrisome proteins with significant change upon aging (FDR < 10%, Fig. 5b, Table S2). Collagen XIV, a collagen of the FACIT family of collagens that is associated with the surface of Collagen I fibrils and may function by integrating collagen bundles³¹, was downregulated in old mice (Fig. 5b). Collagen XIV is a major ECM binding site for the proteoglycan Decorin³², which is known to regulate TGF-beta activity^{33, 34}. Interestingly, our scRNAseq data localized Collagen XIV expression to interstitial fibroblasts, which together with mesothelial cells also expressed Decorin and were distinct from the lipofibroblasts that showed very little expression of this particular collagen (Fig. 5c). Thus, the combination of tissue proteomics with single cell transcriptomics enabled us to predict the cellular source of the regulated proteins, which can be explored in the online webtool. In the webtool the cell type specificity of any gene query can be exported as dot plot in pdf format.’

Below the exported dot plot from the webtool for Collagen IV as an example:

4) The following section indicates that the solubility of proteins does not globally change during aging. The authors identify a few potentially interesting solubility changes with aging (not validated independently). This section a bit disconnected from the rest of the work, needs better motivation and could be better shortened.

Accurate quantification of extracellular matrix changes with aging is a major strength of this work. We have now added an additional Figure on ECM (Fig. 5) to better present changes in total protein quantification of ECM (moved from supplement to main Figure) and use this Figure to present the cell type specific expression of ECM proteins (see above). Thus, the Figure describing solubility changes is now shortened.

5) The following sections on cell-type specific effects during aging reveal potentially interesting insights into how gene expression is selectively changed in certain cell-types. The results in this section show that ~120 genes are differentially expressed in each of the two most abundant cell types with age, thus questioning the authors' previous statement that cell type identity is preserved during aging. This needs clarification.

As the reviewer points out, a careful distinction between cell type identity and differential expression must be made. We used canonical marker genes to manually assess boundaries of cell type identity in the gene expression space. With our clustering method outlined below we found that cell type identity was very little confounded by the aging effects (apart from subtle age dependent shifts from the cluster centers).

We would like to emphasize that we used specific processing to minimize the potential mix-up of cell type identity with age dependent expression changes. As currently written in the Methods section we did not perform the standard highly variable gene selection for independent component analysis and subsequent cell type identification. In the standard approach all expression matrices are first merged and then treated as a single matrix. In such a way one would calculate the highly variable genes across all samples, likely including many genes that change with age between the young and old mice. Instead, we calculated the highly variable genes for each mouse sample independently.

Thus, the highly variable genes determined in such way should represent cell type markers. Next, we used only highly variable genes that appeared in >4 mouse samples for independent component analysis. This approach reduces the risk of incorrect cell type assignment in a data set containing two biological conditions compared to the standard approach. To make this more clear we have added the following sentence into the Results section:

“To ensure that cell type discovery is not confounded by aging effects we only used highly variable genes between cell types (see Methods for details)”

The subsequent validations focus on type 2 pneumocytes. For one MHC class protein, cell-type specific differential expression was confirmed at the protein level using flow cytometry (Fig. 4). Furthermore, Nile Red staining, a proxy for cellular cholesterol levels, is shown to be increased in old mice as predicted based on increased RNA expression in the cholesterol synthesis pathway (Fig. 6). The authors should better explain the nature of the markers used in Figs. 4F, 6D and E, and should clarify how specific they are for alveolar type 2 pneumocytes. Fig. 6: Why were cells stratified there using the forward scatter?

The lineage markers used for FACS in revised Fig. 7 (CD31, Epcam, CD45) enabled us to validate protein levels on epithelial cells only - we did not use a panel which would allow specific conclusions about type-2 pneumocytes. We thus changed the results section text in the following way:

‘...which we validated using an independent flow cytometry experiment on epithelial cells marked by Epcam expression (Fig. 7k).’

We also performed immunofluorescent staining of aged and young mice using LipidTox (#H3447), a compound that stains for neutral lipids, along with anti-prosurfactant protein c (Millipore, AB3786). This data shows that increased LipidTox staining in aged lungs was specific to alveolar type 2 cells. We included the Figure below in revised Figure 8.

The current validations are restricted to one specific example protein (MHC) or the use of an indirect marker (Nile Red). Furthermore, the physiological relevance of these findings is unclear. To show that their resource is indeed useful to the community, the authors must provide a global validation of their predictions for a larger number of genes. I suggest that they use the existing (or a similar) flow cytometry setup (with sorting) followed by global analysis (RNA sequencing and possibly proteomics) to confirm that they can reliably predict cell-type-specific gene expression changes (also in relation comment 3).

As suggested by the reviewer we performed global validation of cell type specific differential gene expression analysis for a large number of genes by flow sorting epithelial cells and macrophages from an additional cohort of young and old mice (see Suppl Fig. S5 for gating strategy) and performed bulk RNA-seq on these isolated cell types (see revised Figure 7). PCA analysis was performed using the scRNA-seq derived signatures of alveolar macrophages and type-2 pneumocytes. Gene expression profiles of flow sorted epithelial cells and macrophages were projected into this PCA space (see methods for details) showing good overlap of cell type identity, thus validating the scRNA-seq based cell type annotation (Fig. 7d, e). Next, age-dependent alterations in the flow-sorted bulk RNA-seq data were identified. Significant agreement with the scRNA-seq derived results was observed (Fig. 7f-j; Fisher's exact test, $P < 2.2e-16$), thus validating the power of scRNA-seq to derive age-dependent changes in gene expression.

Figure 7. scRNA-seq enables cell type resolved differential expression analysis. (d) Scatterplot illustrates PCA of in silico bulk samples of Alveolar macrophages and Type-2 pneumocytes and the projected flow sorted bulk samples. Color and shape indicate cell type identity and data modality. PCA loadings show that well known marker genes define the first principal component corresponding to cell type identity (e). Fold changes derived from the flow sorted bulk samples and the cell type resolved differential expression analysis are depicted on the X and Y axes respectively for Alveolar macrophages (f) and Type-2 pneumocytes (g). The likelihood of corresponding fold change direction was highly enriched between the scRNA-seq and flow sorted bulk data for both cell types (h). X axis shows the odds ratio including 95% confidence interval. Black vertical line illustrates an odd ratio of one representing equal likelihood.

Reviewer #3 (Remarks to the Author):

The manuscript presented by Angelidis et al combines single cell transcriptomics and 'deep' proteomics approaches associated with a panel of bioinformatics and biostatistical tools in an attempt to create a predictive reference map of cellular source of regulated proteins associated with the aging of the lung in a mouse model. It is an exhaustive investigation supported by an exemplary, reproducible and statistically credible experimental design, which results were validated by orthogonal technologies. The respective raw data are available online, as well as the whole lung aging atlas. It is therefore an important contribution not only to the understanding of the lung proteome itself but mainly to the events associated with the aging process in an animal model. In summary, I strongly recommend the publication of this Angelidis et al 's manuscript in the Nat Comm.

We would like to thank the reviewer for taking the time to evaluate our work. We are grateful for the very positive feedback and have edited according to your suggestions whenever possible. The parentheses, which the reviewer refers to as 'typing errors', were intended with the aim to more clearly distinguish R functions from the written text. We have used this formatting before and therefore kept it.

Minors

Abbreviations should be written completely as much as possible when first mentioned in the text.

• Introduction session:

ECM (pag1) . If written in full in the Introduction session then it must be abbreviated in the 'results session'(pag4).

• Results session:

Dropseq (pag2)

tSNE map (pag2)

scRNAseq (pag 6)

Uniprot (pag 8)

GO (pag 8)

KEGG (pag 8)

• Discussion session

CITE-seg (pag 10)

• Methods session

PDMS (pag 17)

UMI (pag 17)

There are some 'typing errors' such as () in the Methods session that need to be fixed, for example: pg 18: Seurat's NormalizedData(); scaleData(), et, etc....

Figure 6C: Please increase the letter size of the cholesterol biosynthesis pathway to make it readable.

Reviewer #4 (Remarks to the Author):

Angelidis and colleagues present an atlas of lung cells from aging mice. Moreover, they provide a link of single cell data to bulk proteomics data with excellent correlation between RNA and protein data strongly supporting the enormous value of single cell RNA-seq data. Furthermore, as an atlas resource the Theis lab provides a web interface for these data and also link the data to other atlas efforts such as the mouse cell atlas. As a resource paper, it is expected to stay mainly descriptive, albeit some attempts have been made to validate some of the work at least phenotypically. With the enormous expertise of the group of authors and laboratories, it is not surprising that the manuscript is very well presented. Nevertheless, since all atlas efforts are critically shaping the landscape of how cells are described during the next decades, several points are to be addressed first.

We would like to thank the reviewer for taking the time to evaluate our work. We are grateful for the very constructive criticism and addressed your concerns as outlined below.

1. *Albeit ~15,000 cells are a significant number of cells, it is still on the low site for a full organ in the mouse. Particularly, since this is the number of cells for both young and old animals. Moreover, the two major cell types, alveolar macrophages and type 2 pneumocytes make up a substantial portion of the overall number of cells, which leads to very few events for rare cells in the dataset. Furthermore, the authors correctly cite a paper from 2008 stating already 40 cell types, yet the authors only discover 30 cell types. Three things need to be added to the manuscript. One, a comparison of the 30 cell types with the 40 previous cell types and a reason why the other once might not have been detected in this new study. In other words, why are we falling behind current knowledge about existing cell types. Second, additional cells should be added which specifically might add to getting closer to the previously described 40 cell types. Third, a general remark in the discussion section, how such atlases are going to be build up further in the future by e.g. adding additional cells later to the dataset. The reader needs to understand the dynamics of such atlases, the current limitations (cell number, sequencing depth, any technical biases) and how this will be solved in the future.*

Definitions of cell type identities are often very vague. The cited 40 cell types are derived from literature meta-analysis of mouse and human studies and include not well defined definitions of cell types. We believe that unbiased single cell analysis will eventually come up with a more correct number and rather represents ground truth than literature meta-analysis. Furthermore, the intention in this study was not to establish a comprehensive atlas of all possible cellular states in mouse lungs, but we intended to perform an analysis of aging effects at a depth of current state of the art. Therefore, we cannot conclude that we 'fall behind' existing knowledge with this study. We want to emphasize that 1) in the current data we were able to identify even very rare (< 0.1%) cell types (ie megakaryocytes) and 2) comparable scRNAseq data sets such as the MCA also 'only' identified 32 cell types in adult mouse lung. Even though it is possible (if not likely) that we might discover additional very rare cell types with higher cell numbers, we conclude that the current depth of this study on lung aging provides an overview to cell type identity which is comparable to the current state of the art single cell resources 'Mouse Cell Atlas' and 'Tabula Muris'.

We changed the discussion of the Atlas aspects in the discussion section to the following sentences:

‘In this work, we present the first single cell survey of mouse lung aging and computationally integrate single cell transcriptomics data with bulk proteomics and transcriptomics of whole lung to build a first draft of an atlas of the aging lung. Atlasing efforts are generally organized in stages so that more detailed maps of cellular phenotypes will be integrated at later stages to initial drafts of the atlas. The intention in this study was to perform an analysis of aging effects at a depth of current state of the art.’

2. Material and Methods Section: The authors describe red blood cell lysis as a step of generating single cell suspensions. How did the authors quantify contamination with red blood cells, and therefore with blood? How do the authors exclude that the remaining cells do not have any blood-derived contaminations? What about the immune cell compartment (e.g. T cells, B cells, monocytes). Is there a comparison with blood derived immune cells to see differences between the tissue-resident immune cells and blood-derived immune cells? These are important technical issues, particularly for the lung and require further explanation. Without any statement about quantification of the potential contamination of blood-derived cells, the lung atlas might be a mixed lung-blood atlas. The reader and user of the atlas needs this information.

The reviewer makes an important point about potential blood-derived contamination. To quantify blood-derived contamination we performed thorough comparison with both peripheral blood and lung data from the MCA. We observed high matchSC scores when comparing our cell type signatures with the MCA lung data. However, when comparing our cell type signatures with the MCA peripheral blood data matchSC scores were much lower with the exception of one cluster in our dataset which represents red blood cells. Of note, all but one mouse in our dataset had very low frequency of that red blood cell cluster (see Suppl Fig. S1d). For the purpose of this analysis the red blood cell cluster serves as a control since these cells are only found in blood and exemplify the matchSC score that can be expected when true cell type overlap exists even across studies and technologies.

We now describe this analysis in the Results section:

“Moreover, when comparing our cluster identities to the MCA peripheral blood data only weak correspondence was observed (Fig. S2b), which was similar in the MCA peripheral blood versus MCA lung comparison (Fig. S2c). One notable exception in this comparison is our cluster of red blood cells which achieved high correspondence with the MCA peripheral blood cluster annotated as “Erythroblast_Hbb-a2_high”. The red blood cells serve as a control and illustrate matchSC values for a correct overlap (Fig. S2d). Taken together these findings indicate that very little blood-derived contamination was present.”

Additionally, we have generated a new supplemental figure to illustrate this analysis:

Figure S2

“Supplementary Figure S2. Comparison with the Mouse Cell Atlas validates lung cell identities. (a-c) The matchSC score comparison between the clusters in this study, the MCA lung and peripheral blood signatures is shown. Red and white colors indicate high and low matchSC scores, respectively. The outlier in panel c represents red blood cells (purple rectangle). (d) The box plot shows the distribution of maximal matchSC scores for each cluster across the comparisons between these three data sets. The outlier in the comparison between cell types in this study and the MCA blood data represents red blood cells (underlined in purple).”

3. *The age-related changes of the major cell types is a major asset of the manuscript. In alveolar macrophages, Marco as an altered cell surface molecule is highlighted. Similar to the alveolar type 2 pneumocytes (flow cytometry for MHC) should be added for Marco on alveolar macrophages.*

We tried to validate Marco by flow cytometry and unfortunately could not get the staining to work in our FACS panel so that we could not evaluate Marco protein levels on alveolar macrophages. Since the significance of RNA-seq differential gene expression of flow sorted macrophages was not good (even though the direction of regulation was the same as in the scRNA-seq data -see figure below) we decided to remove the sentence about Marco in the results section.

4. *Figure 1: It is not entirely clear from the main result text, how the authors named the cells in Fig 1d (it is written in the methods section, but could be presented more prominent, in the end this is what a cell atlas does). For some cells (e.g. dendritic cells), it would be good to use updated nomenclature or at least add this nomenclature. What about subtypes of known cell types? For example, some of the immune cells identified are known to exist in subtypes. Explain, why you did not add this level of information. Or alternatively, point out that this is available on the website.*

The clusters in Fig. 1 B cannot be linked to the cell nomenclature used in Fig. 1c and d. This is very difficult for the reader to follow. What are Fn1+ macrophages in context of the existing literature?

We did not add more information about subtypes as we are limited in numbers and resolution to really be sure about these (Aging atlas draft 1.0). As stated in our comment to point (1) the intention in this study was not to establish a comprehensive atlas of all possible cellular states in mouse lungs, but we intended to perform an analysis of aging effects at a depth of current state of the art for the main cell types we were able to annotate. In some instances (e.g. Ccl17+/CD103-/CD11b- dendritic

cells) we annotated cell clusters with marker genes showing highest fold change and/or lack of important markers present on related clusters. In panel c of Figure 1 we have ordered these cell annotations by similarity (dendrogram from correlation of marker signatures), showing that we already pick up quite a few subtypes of DCs, macrophages, etc. Annotations in panel c are labeled with both cell type names and the cluster number from panel b so that actually it is possible to link the two panels. The Fn1+ macrophages are currently to our knowledge not described as a distinct subset in the existing literature. We are following up on these cells and will better characterize the developmental origin, location, identity and function of this new cell population in future work.

5. Figure 2: The RNA-seq and protein data are strikingly correlated, except for Collagen IV. Is there any functional or molecular explanation for the different behavior, while all other systems/pathways were correlated? If you do not have any proof, can you hypothesize?

The abundance of mRNA and protein can be decoupled due to posttranscriptional regulation of protein synthesis/turnover. Increased accumulation of Collagen around airways and vessels in aged mice may be due to decreased proteolytic turnover of type IV collagens. We may speculate that a feedback signal exists that reduces transcription of the very same collagens if they accumulate on protein level.

6. Figure 2 and S3: Age dependent alterations of relative frequencies: How did the authors ensure that there was no technical reason for differences in relative frequencies? Did the authors control their relative frequencies with another independent single cell methodology? If not, such information should be added.

FACS validation of relative frequency differences would not be effective as the same biases for isolation would be present. Bulk tissue RNA-seq however enabled us to additionally validate the change in cell proportions in whole lung by deconvolving the bulk expression data using our single cell gene expression profiles (see revised Figure 4). Indeed, we found that the ciliated cell marker genes signature was significantly upregulated in old lungs (Fig. 4e/f in revised manuscript). For *in situ* validation we counted more than 2000 cells from a large number of airways and several mice. Of note, with this deconvolution method we also independently validated the increased frequency of several immune cell populations (T cells, monocytes, ..) that was also evident in the scRNA-seq data.

Figure 4

Figure 4. Whole lung cell type deconvolution reveals increase of Ciliated cells in airways of old mice. (a) The MDS plot shows the mouse-wise euclidean distances of cell type proportions for the two age groups (b) The box plot shows the significant difference in the multidimensional scaling component 1 of cell type proportions between young and old. (c) The Fruchterman-Reingold (FR) embedding of the airway epithelial cells in the dataset reveals distinct clusters of airway cell identity. (d) The indicated color code shows the distribution of young and old cells to the three clusters presented in panel (c). Note the increased density of old cells in the Ciliated cell cluster. (e) The volcano plot shows negative log₁₀ enrichment p-values of cell type marker signatures in the differential expression results of the bulk RNA-seq data from young and old mice. (f) The empirical density plot shows significant enrichment for Ciliated cell type marker genes (red line) compared to all other genes (black line) in the distribution of fold changes derived from the bulk differential expression analysis. (g) Club and Ciliated cells were stained using a CC10 and Foxj1 antibody respectively. (h) The boxplot depicts the quantification of Ciliated cells from counting a total of 2647 Club and Ciliated cells in 14 individual airways of two mice of each indicated age group. (i) Ratio of Ciliated to Club cells in 14 individual airways of two mice for each indicated age group. P-values are derived from an unpaired, two-tailed t-test using Welch's correction.

7. Figure 3: "Indeed, we were able to confirm the mesothelium specific expression of *Fras1* using our scRNAseq data". How is this statement supported by the visualization in Fig. 3j/k?. Fig. 3l is

mentioned in the text, but does not exist. The information for Decorin is not included in Fig. 3j/k. In general, the authors need to make sure that figures, gene names and genes/proteins mentioned in text better match.

The only point for these panels was to illustrate the power of scRNAseq to identify cellular sources of proteins with the example of Col14a1 expressed by interstitial fibroblasts. Since the additional sub-clustering of mesenchymal cells did not really strengthen this aspect we removed that analysis and instead show now expression specificity of Col14a1 and its binding partner Decorin in the main data object (described in Figure 1). The new panel can be found in revised Figure 5.

“Figure 5. scRNA-seq predicts cellular origin of age-dependent protein alterations. ... (c) The dot plot shows mRNA expression specificity of Col14a1 and its binding partner Decorin (Dcn) in the scRNA-seq data.”

8. How many cells came from old, how many cells from young mice? One can infer from the text and the supplement, but it is not explicit. Add in main result text.

We added the number of cells that came from young and old mice into the Results:

“After quality control, a total of 14,813 cells were used for downstream analysis (7672 young, 7141 old).”

9. First sentence in discussion not necessary, it is a political statement. Increasing health, probably everybody would agree. Increasing lifespan is a very much debated statement. Enabling healthy aging might be ok again. Re-phrase.

We changed the sentence to: ‘Enabling healthy aging is one of the prime goals of the modern society.’

10. Page 9: sentence of main text: This is a statement that is too negative and should be avoided. In This paper should stay away from the old discussion whether RNA could be used instead of protein. As long as proteomics cannot provide single cell data to a similar extend as scRNA-seq is doing it at the moment, these statements are not helpful.

We understand the reviewers point. However, the reason to bring that up in the discussion is that we can clearly observe decoupling of mRNA and protein (as prominently shown in Figure 2) and thus want to emphasize the importance of developing multi-omics single cell methods. We anyway also illustrate in our paper that mRNA and protein methods are significantly correlated overall and in particular on pathway enrichment level. We therefore only slightly adapted the sentence in the discussion to make it less negative.

‘The example of basement membrane collagen IV genes that were all downregulated on the mRNA level but upregulated on the protein level illustrates that protein posttranscriptional regulation is important. In particular the abundance of ECM proteins, which often have long half lives and are thus likely more often regulated on the posttranscriptional level could frequently show decoupling of protein and mRNA.’

11. Page 10, first paragraph: citation still in doi-format

Thanks for pointing that out.

12. Bioinformatic processing of scRNA-seq reads: Please provide some more statistics concerning the quality of the data. How many barcodes were detected, how many made the chosen cutoff, how many UMIs and genes per cell were recovered, mean read length, etc. Provide as a supplementary figure.

We agree with the reviewer that additional quality statistics would be helpful. Therefore we have added a number of panels to Figure S1 illustrating the fraction of cells per mouse and cell type, the distribution of total UMI counts and genes per cell for each mouse and the mapping statistics including percent mapped, average read length and total reads.

“Supplementary Figure S1. High technical reproducibility enables integration of the 15 mouse experiments. (a, b) The violin plots show the distribution of the (a) number of genes detected per cell and (b) total UMI counts per cell across mice, respectively. (c) scRNA-seq alignment statistics show comparable values across mice. (d) Cell type identity and the fraction of cells per mouse are shown on the X and Y axes respectively. (e, f) tSNE visualization colored by (e) mouse sample and (f) age group.”

13. Data availability paragraph requires addition of the respective numbers instead of XXX.

All raw data will be uploaded to GEO and PRIDE upon final submission of the manuscript.

REVIEWERS' COMMENTS:

Reviewer #1 (Remarks to the Author):

The authors have done a remarkable and thorough job in addressing all of my comments. I have no further concerns.

Reviewer #2 (Remarks to the Author):

The authors have fully addressed all my comments. The analysis and validation are clearly improved compared to the previous version. I therefore recommend publication.

Reviewer #4 (Remarks to the Author):

In this revised manuscript the authors have addressed all major questions raised by this reviewer.

Very minor concerns left:

1. Adding additional information resulted in an incorrect order of the subpanels in Figure 7
2. In Line 151 it probably must be Fig.3a instead of Fig.2a)
3. In Lines 195ff it probably must be Fig. 4 several times instead of Fig. 3

Reviewer #4 (Remarks to the Author):

In this revised manuscript the authors have addressed all major questions raised by this reviewer.

Very minor concerns left:

1. Adding additional information resulted in an incorrect order of the subpanels in Figure 7
2. In Line 151 it probably must be Fig.3a instead of Fig.2a)
3. In Lines 195ff it probably must be Fig. 4 several times instead of Fig. 3

All changes have been made according to the reviewers comments.